

# A New Propose for Prehistoric Tritonis Lake's Location based on Apollonius of Rhodes' description

Stavros Papamarinopoulos[1], Panagiota Preka-Papadema[2], Konstantinos Kalachanis[3][4], Habik- Hasak Maroukian[5] Georgios Saraditis[6], Dimitrios Theodosopoulos[6],Chris Tzanis[7]

[1]University of Patras, Department of Geology, 26504 Rio Patras, Greece
[2]University of Athens, Department of Astrophysics, Astronomy and Mechanics, Faculty of Physics, 15784 Athens, Greece
[3]New York College,10558, Athens, Greece
[4]Aristotelion School of Korinthos, 20100,Isthmos, Korinthos, Greece
[5]Department of Geography and Climatology, Faculty of Geology and Geoenvironment, National University of Athens ,
Athens, GR-15784, Greece
[6]Independent Researcher
[7]Climate and Climatic Change Group, Section of Environmental Physics and Meteorology, Department of Physics, National and Kapodistrian University of Athens, 15784 Athens, Greece

*Correspondence to*: Konstantinos Kalachanis (kalahanis@hotmail.com)

**Abstract.** The Argonauts' trip along the northern limit of the Libyan Desert, from Syrtis Gulf to Tritonis Lake, is described by Apollonius Rhodes' text '*Argonautica*'. The Argonauts sailed in this lake and located a narrow passage through which they exit to the Mediterranean's coast. In this work, we followed the ancient text carefully, step by step, in order to find the 'unknown' location of pre-historic Tritonis Lake. All the steps of the trip were tested trans-scientifically and found correct in

many aspects. Today's many wadis, sabkhas and oasis, which cover the northern part of the Libyan Desert, are the remnants of numerous older rivers or lakes existing there in the 13th century BC, when the Argonautic Campaign happened. The current lakes of the Siwa Oasis were part of the Tritonis Lake, which was extended to the east and covered a large part of today's Qattara Depression. Using modern technology, we found the narrow water passage which led the Argonauts from the Tritonis Lake to the Mediterranean Sea, near El Alamein's coast. From there, Argo sailed eastwards parallel to the African

coast till a promontory, according to Apollonius. This is the Abu Qir Cape in the Canopis Nile Delta. From there, the ship reached eastern Crete, since the sailors had previously seen the mountains of Karpathos Island, as Apollonius notes.

## 1 Introduction

The Tritonis Lake, mentioned in the ancient Greek literature, was located in a region of the North Africa. Its name derives from the sea deity, Triton which was the personification of reciprocal coastal wave**.** Pausanias (*Greciae descriptio,* IX, 21, 1,

2–IX, 21, 2, 1) describes him to have thick deep green hair (shallow sea color), with a large mouth (shore length), large white eyes (coastal foam), large fingers and toenails (traces of the retreating coastal wave on land) and from the sternum down dolphin tail (as a rush coming out of the sea). This phenomenon is intense in the Nile's delta and according to



History of
Open Access Geo- and Space
Sciences
Discussions

Apollonius of Rhodes, 'Triton' is the oldest name on the Nile River (Argonautica, IV, 259-260 &267-271): '*For there is another course, signified by those priests of the immortal gods, who have sprung from Tritonian Thebes….in the days when*

*Egypt, mother of men of an older time, was called the fertile Morning-land (Ἠερίη) and the river fair-flowing Triton, by which all the Morning-land is watered; and never does the rain from Zeus moisten the earth; but from the flooding of the river abundant crops spring up.'* Luxor which is located on the banks of the Nile River is the ancient '*Tritonian' Thebes* (see also Hecataeus *Fragmenta* fr. 18a, line 2).

The Argonautic campaign, which is connected with the Tritonis Lake, must have taken place some decades before the
Homeric Trojan War, because the participants of the Argonautic campaign, such as Telamon and Peleus, were fathers of the heroes who fought in the Trojan War as described by Homer. Achilles was a toddler bids and he farewell to his father Peleus (Orpheus, Argonautica, 395-399; Apollonius, Argonautica, I, 94). Also, Hercules, which was synchronous with the Argonautic campaign, he participated in the 'first' Trojan War against King Laomedon, father of Priam. According to Apollonius, Hercules performed his 11th labour, in Tritonis Lake, by grabbing the apples of Hesperides (Apollonius,
*Argonautica*, IV, 1396-1450). Additionally, according to Apollonius, in Tritonis Lake, the Argonauts killed a shepherd named *Κάφαυρος (*Caphaurus), who was a descendant in the 3rd generation of the King of Crete, Minos, as the grandson of his daughter, Acacallis. However, the King of Crete, Idomeneus, who is also descendant of King Minos in the 2nd generation, had participated in the Trojan War (Homer, Iliad, II, 645). However, the Homeric Trojan War lasted between 1227-1218 BC, according to Papamarinopoulos et al. (2014). Other views project the fall of Troy to have been conducted about 1250 BC or
1300 BC (Papamarinopoulos et al. 2012). Consequently, the Argonautic campaign was conducted *in the 13th century BC, with Tritonis Lake already existing at that time* in North Africa.

However locating this lake is unclear due to confusing references provided by various (ancient and modern) authors. It seems that after 1300 BC the lake has been dried up or in any way changed and the authors have difficulties to agree to its location. Diodorus of Sicily claims that an earthquake caused the extinction of the lake (Diodorus of Sicily, III, 55, 3,
10).The *seismicity* of the North Africa region is well known, as a result of the relative motion of the African and Eurasian plates. According to Nur (2000), an intense seismic activity had occurred in the Eastern Mediterranean during the 13th to 12th centuries BC. Also, Herodotus (IV, 173) describes an arid period *'dried up the water-tanks, and all countries, lying in the region of the Major Syrtis, were waterless'.* This aridity is related to 'Nasamones' descendants of 'Nasamon', brother of Caphaurus (Apollonius, *Argonautica,* IV, 1490-1497). Both of them were 3rd generation descendants of King Minos. This
means that this dry period occurred after the Trojan War, *during the dark ages.*

Kaniewski et al. (2010) suggest that the decline of several civilizations in eastern Mediterranean, which taken place during the so-called *dark ages,* happened during the first quarter of the 12th century BC, due to *climatic changes*. Langgut et al. (2013) assumed that Middle East and North Africa region experienced a drought that lasted between 1250 and 1100 BC, which had a negative result in the economies of countries like Egypt and Kingdoms of the Mycenaean World. Several
authors have reached similar conclusions (e.g. Kaniewski et al., 2008; Bernhardt, et al. 2012, Drake, 2012, Finné et al., 2017,

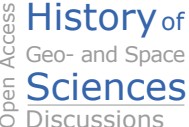

Finkelstein et al., 2017). Thus, a climate change after ~1300 BC, in the region of North Africa, resulted to a decreased lake water level, to such an extent that it eventually ceased to exist in historical times.

However, the authors who refer to Tritonis Lake connect it geographically with the gulf of Syrtis, where Argo had strayed (*Orphica Argonautica* and *Apollonius Argonautica*). It is known that in both Syrtis, the navigation was problematic. Pliny, in
*Natural History* (5.26), mentions that '*the gulf was formidable because of its shallow and the tidal water of both Syrtis'*. Strabo  writes [17.3.20]: *'the difficulty with both the Major and the Minor Syrtis is that in many places the water is shallow, and at the rise and fall of the tides ships sometimes fall into the shallows and settle there, and it is rare for them to be saved'*. In the text of Apollonius, there is a detailed description of the dangerous conditions of that region (IV, 1223-1276): *'Where from is no return for ships, when they are once forced entering into that gulf. Because there is a swamp everywhere, algae*
*come up from everywhere, from the bottom of the sea, and above them, the brilliant spray of the noiseless waves rises. The sand reaches up to the horizon line and in this area no reptiles or bird live. The tide dropped the ship to the deepest point of the coast..... What a desert area is this that seems in front of us!* ... '*We must suffer the cruelest woes, having fallen on this desolation! ... But now the tide rushes back to the sea, and only the foam, whereon no ship can sail, rolls round us, just covering the land'.* This detailed description fits perfectly in the environment of the Major Syrtis which is in front of the
Libyan Desert. The Minor Syrtis is in Tunisia, near to the Mountain Atlas' area. Also, Argo drifted by a north wind and reached the gulf of Syrtis, coming from the Ionian Sea, which is just north of the gulf of the Major Syrtis.

According to Apollonius *Argonautica* and Pindarus' text *(Scholia InPindarum, Scholia in Pindarum (scholia vetera)* Ode P 4, scholion 46, line 4), the Argonauts, *coming from the gulf*, carried the ship for 12 days and nights *in the desert of Libya inland* and they finally reached Tritonis Lake. Also, this lake is located *in the middle of Libya*, according to Dionysius
Periegitis (Orbis description 267-270*)*. However, Strabo (XVII, 3, 20, 39-43) locates Tritonis Lake in Pseudopenias cape in the city *Berenice*, today's *Benghazi*, near Major Syrtis Gulf, in Cyrenaica peninsula. Moreover, Clendenon (2009) taking into account Apollonius' description and accepting the presence of the Argonauts in the Major Syrtis Gulf, writes: 'I speculate that the Argo initially may have grounded near the modern-day port of *Ras-Lanuf* with shallow offshore waters and rocks….then the modern-day *Bight of Brega* could have been the bay to which the Argonauts prayed to be led. … *Perhaps*
*Tritonis Lake approximately coincided with today's Sabkha Chuzayyil. ...* Then, the coastline changed, probably moving northward …Consequently today's Sabkha Chuzayyil is situated inland from the gulf... the single navigable gap between Tritonis Lake and the open sea may have been either a submerged wadi valley (small canyon, a gully or ravine that usually is dry but becomes a torrent after heavy rains) or a cleft in the underwater rocks…. Today these extinct wadis must be covered by windblown sand and at higher relative sea levels and therefore lost to human view'.

A different location of the Tritonis Lake is given by Herodotus. He clarified that *Triton River and Tritonis Lake is near to the land of Lotophagoi,* who were located in the islet Meninx in the gulf of Minor Syrtis (IV, 178, 3-7): '*Next to these along the coast are the Machlyes, who also use the lotus, but less than the aforesaid people. Their country reaches to a great river called the Triton, which empties into the great Tritonian Lake, in which is an island called Phla'.* Also, according both,


Dionysius Schytovrachion and Diodorus Siculus, the *Ethiopians* lived close to the Tritonis Lake and the Ocean laid a big

mountain named *Atlas (*Dionysius Scytovrachion, *Fragmenta,* 1a, 32, F, fragment 7, line 44*).* The above descriptions suggest the *chotts of Tunisia* to be the river Triton and Tritonis Lake. Chott is a lake that stays dry through the hot season but contains some water in winter. It receives water from small creeks during the rainy season, but has no natural outlet. Clendenon (2009) mentions that 'modern scholars echo one another' placing Tritonis Lake in this region and identify it with the modern-day *Chott el Djerid (chott el Jerid).* However, the same author clarify that the Tunisian location of the lake is

rejected by Buxton (2004).

Moreover, Herodotus (IV, 179) mentions the rumor that after Argo was constructed and *before Argonautic campaign* was conducted, Jason sailed from Iolcus to the oracle of Delphi bypassing Peloponnesus (since Corinth's canal did not exist), in order to transport a bronze tripod as a tribute. When Argo reached cape Maleas of Peloponnesus, a strong north wind dragged her in Minor Syrtis' region, near the land of Lotophagoi. It is exactly the same route that was later followed by

Odysseus (*Odys.* IX, 80-84). Argo was found 'on the rocks of Tritonis Lake', when 'God Triton' appeared asking the tripod in order to show them the narrow passage leading them back to the sea. We note that a similar description is presented by Apollonius (*Argonautica*, IV, 1537-1619) when Argo, *during the Argonautic Campaign*, arrived in Tritonis Lake and then it helped by 'God Triton' to came back to the sea, through a narrow passage, after offering him the sacred tripod. It is noticeable that Herodotus referred to *another Argo's trip* and its arrival into Minor Syrtis. If the ship, during Argonautic

campaign, had arrived in Minor Syrtis too, Herodotus would certainly refer to that as well. Additionally, the location of Triton River in Tunisia *is in contrary* with the Apollonius' reference that the *Triton River* is the *Nile River*, in Egypt. Although Herodotus is a historian, Apollonius has been director of the Alexandria's Library and he had direct access to multiple bibliographies.

Consequently, the question of the prehistoric Tritonis Lake's location remains. In this work, we will try to answer where

exactly Tritonis Lake was located based on the detailed description of *Apollonius Argonautica* text. The purpose of this study is the location of the position of a large water basin in North Africa (Libya-Egypt) which existed during prehistoric times and 'disappeared' in historical times for an unknown reason (climate factors, geological changes etc). Any connection there is between this great lake with remaining waters (sabkhas, wadis etc) and the present land formation of these areas is significant to geophysical research. Finding the 'correct' location will lead to on-site ground research that will produce useful

findings for understanding the palaeoclimatology and palaeogeology of the region which today is 'dry'.

## 2 Locating Tritonis Lake's exit to the Mediterranean Sea

The Argonauts, from the coast of Africa, arrived in East Crete (*Orphica Argonautica,* 1355-1362 and Apollonius *Argonautica,* IV, 1636-1690), in the gulf of Dykti, (today's gulf of Saint Nicolas). The next day, having departed from the gulf (Apollonius Argonautica, 1690-1720) arrived in the north-eastern edge of Crete, in the Cape of Salmonis (Fig. 1(left)).

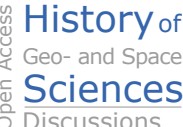

From there, they sailed through to Cretan Sea in order to reach to Anafi Island (e.g. Kalachanis et al, 2017). Both ancient authors have depicted sailing in the Cretan Sea as extremely dangerous (Orphica Argonautica IV, 1364-1368 and Apollonius *Argonautica*, 1704-1730). Argo was saved only because of God Apollo's help and the Argonauts built in Anafi Island, *an initial altar* in honor of Apollo which, after many years, became a great temple. Its ruins are still present in the southern part of Anafi Island.

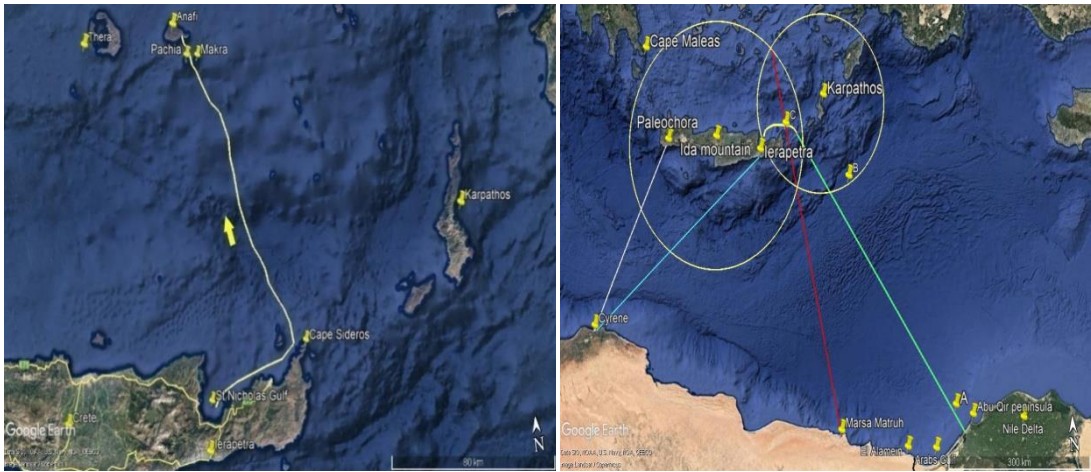

*Figure 1 (Left) Argo arrived in the gulf of Dykti (Saint Nicolas), in East Crete. Cape Sideros (or Cape Salmonis) exist at the eastern end of the island. From there, the Argonauts sailed through the Cretan Sea and reached to Anafi Island. (Right) The two circles represent the distance from which the mountains of Karpathos or Crete could be seen with naked eye. This distance is close to 125 km from Lastos Mountain of Karpathos Island (1215 meters) and 178 km from the Ida Mountain of*

*Crete Island (2456 meters) respectively. The intersected circles are defined with (red) line. The latter as projected southwards, meets the Egyptian coast, in Marsa Matruh. The Argonauts' route (green and yellow lines) from African coast (Abu Qir Cape) to East Crete (Saint Nicolas Gulf) is shown. The distance between Cyrene and West Crete (Paleochora) is 305 km while the distance between Cyrene and East Crete (Ierapetra) is 418 km. © Google Earth*

Consequently, this route, provided by both texts, is confirmed by the archaeological findings of the temple of Apollo, in

Anafi Island. However, Argonauts, when they were in Tritonis Lake, asked from 'God Triton' to show them, the direction they should sail in order *to approach the Peloponnesus and no to reach in East-Crete Island* (Apollonius Argonautica, IV, 1564-1570), because they intended to follow the east coast of Peloponnesus, heading north, in order to reach Iolcus (Fig.1 (right)). Indeed, after Anafi Island, Argo *traveled westwards and reached Peloponnesus,* either north (*Aegina Island)* or south of it (*Cape Maleas),* according to Apollonius *Argonautica* and *Orphica Argonautica,* respectively.

Additionally, the ancient writer gives very important information. Argonauts, sailing away from the African coast, saw from afar first the '*mountains of Karpathos Island*': *'And rugged Karpathos far away welcomed them; and thence they were to*



*cross to Crete, which rises in the sea above other islands. ... when they came to the roadstead of Dycti' s haven'*. Karpathos is a small island which is located northeastern of East-Crete (Fig. 1). The highest top of Karpathos Island is in the *Lastos Mountain* (1215m). The highest top of Crete Island is in the *Ida Mountain* (2456 m). According to the formula of the *felt*

*horizon*, $3,6 \cdot \sqrt{heigth}$ , we can calculate the distance from which any mountain will become visible to a ship, ([http://www.geo.auth.gr/courses/gge/gge322y/chapter031.html](http://www.geo.auth.gr/courses/gge/gge322y/chapter031.html)), with a sailor's naked eye. The 'height' in the formula corresponds to the height of the mountain. Consequently, the mountains of Karpathos could be visible, when the ship was close to 125 km, while the mountains of Crete would be visible from a distance of 178 km, respectively (see the cycles in Fig. 1 (right)). Obviously, if a ship travels from any location of West and Central African coast (up to Cyrene) to Crete

Island, it will first see the high mountains of Crete, being 178 km away from the island. Karpathos's mountain being East and slightly North of Crete *cannot be seen at all*.

In the Introduction, we presented the proposed locations of Tritonis Lake close to Minor Syrtis or to Major Syrtis or Benghazi in Cyrenaica Peninsula. However, if Tritonis Lake was in Minor Syrtis, following the northward route pushed *by a south wind*, as proposed by Apollonius (IV, 1627-1637), Argo would have arrived to Sicily and not to Crete. An alternative

route, could be sailing from Minor Syrtis to Cyrenaica peninsula and then northward to Crete. If Tritonis Lake was in Major Syrtis, Argo would move along the coast until Cyrene to Cyrenaica peninsula and from this location would sail northwards to Crete. Thus, in any case, Argo would reach Cyrenaica Peninsula, *near to Cyrene*, and then would sail directly *north*, assisted by the *south wind,* as Apollonius described. However, in this route, the sailors are impossible to see first the Karpathos's mountain, as they are heading towards Crete Island.

Additionally, following this description, Argo will reach *West and not to East Crete* (Fig. 1 (right)). Indeed, the closest approach between North Africa and Crete Island is the distance Cyrene - Palaeochora port, Chania (*only 305 km*). This distance would allow Argo, with an average speed about 6-7 km/hour (Casson, 1951), to reach *West Crete*, in 1.8-2.1 days and then they could sail easily westwards to cape Maleas, in Peloponnesus (Fig. 1 (right)). The only case that Argo, coming from Cyrenaica peninsula, could approach East Crete is the change of the south to south-eastern direction of the wind. Thus,

Argo would have drifted toward south-eastern Crete, where there were many ports, in antiquity. However, if Argo had arrived in one of them, it would be reasonable to follow a coastal route along south Crete going *westwards* in order to approach Peloponnesus. The sea route within the Cretan Sea was known from antiquity and continues to be even today notoriously difficult and risky. Another case is the Argonauts to sail along the African coast reaching Egypt and then travel northwards to East Crete, trapping themselves in a time consuming and risky long trip, without any reason.

Summarizing, we conclude that *the Argonauts did not sail from Cyrene to East-Crete* but from another location of the North African Coast. In other words, the *ancient Tritonis Lake* was not located in the entire West and Central African Coast including Minor Syrtis, Major Syrtis or Benghazi as proposed by various authors.

We examine which geographical coordinates in the African coast could be the 'starting point' of the voyage to East-Crete, taking into account that Apollonius' mention that the Argonauts saw firstly the Karpathos's mountains. The intersected

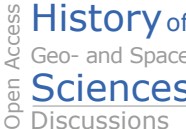

circles of Fig. 1 (right) are defined with the red line. The latter as projected southwards, meets the Egyptian coast, in *Marsa Matruh*. The coastal region of *Marsa Matruh*, with coordinates 31.3543° N, 27.2373° E, is in the *same meridian* with Karpathos Island (35.5066 ° N, 27.2124 ° E). Thus, the only case for seeing *firstly* the Karpathos' mountains, by the Argonauts, is that the ship traveled from any coastal point, e*ast of Marsa Matruh.* It is known that the Cretans and the Egyptians had many shipping contacts even earlier that the 13th century BC. Consequently the voyage Egypt-Crete was known to the Argonauts.

We attempt to locate the Egyptian coast from which Argo sailed to East Crete, based on Apollonius' description (IV, 1578-1585 and 1622-1628): ....'*but hold to the right , when you have entered the swell of the sea from the lake, and steer your course hugging the land, as long as it trends to the north; but when the coast bends, falling away in the other direction, then your course is safely laid for you if you go straight forward from the projecting cape'... 'But at dawn with sails outspread they sped on before the breath of the west wind, keeping the desert land on their right. And on the next morn they saw the headland and the recess of the sea, bending inward beyond the jutting headland. And straightway the west wind ceased, and there came the breeze of the clear south wind; and their hearts rejoiced at the sound it made'.* Argo's eastern sail along the coastline was assisted by the favorable west wind. After 24 hours of continuous sailing from the point of Argo's departure along the coast, the Argonauts reached a promontory. From that cape, the ship left the African coast with the assistance of favorable south wind, with destination the East Crete.

It is important to note the differentiation between the point on the Egyptian coast where Argo arrived from Tritonis Lake (*arrival point)* and the point of departure *(departure point)* as Argo abandoned the African coast. Between these two points *the coast is directed northwards,* exactly as it happens after Marsa Matruh and particularly eastward from the *Arab's Gulf* (Fig. 1 (right). The Argo's *departure point* is near to a promontory. East of the Arab's Gulf (Fig. 1(right)), there is a promontory named *Abu Qir* and a deep gulf after it, in the Canopis branch of Nile's Delta, exactly as Apollonius described it. It is noticeable that, the narrow strip of land located between Moreotis Lake (near to Alexandria) and the Mediterranean Sea consists of *non-alluvial formations*. The coastal zone between El Dabba (near to El Alamein) and Abu Qir peninsula is composed of two Pleio-Pleistocene calcareous ridges composed of chalky, marly, arenaceous and oolithic beach materials having in between recent lagoon all deposits (subkhas), (Hassouba1995; Ibrahim & Mansour 2002). This means that this coastal environment has been stable for the Quaternary period and has undergone minimal changes.

The Argonauts, from the promontory, turned their course towards the open sea (Fig. 1 (right). According to Apollonius, the Argonauts being in the course of the straight line (AB in Fig. 1 (right)), they saw from afar the Karpathos Mountains, from the point B. The route from Abu Qir Cape up to this point is equal to 421 km. Also, the distance BC until the Dikti's gulf is 206 km. Argo traveling with an average speed of 6-7 km/hour covered the whole distance of 627 km in 3.7-4.3 days and nights. We note that, Canopus branch of Nile River named after *Canopus, the captain of Menelaus' ship*, who died there, after the fall of Troy (Od. 4, 351-586). The connection between *Argo and Canopus* occurred in the night sky. The brighter star of the constellation Argo, and one of the brightest stars of the sky, is named Canopus. The ancient Greek writers who



were giving names to the stars and constellations, much later, they deemed it appropriate *to link* Argo's arrival in the place named Canopus, because this place marked the essential end of their tribulations.

According to Apollonius, the time span of the voyage between the *'arrival'* and *'departure'* points was 24 hours. Consequently, the distance between them is about 144-168 km, taking into account an Argo's average speed about 6-7 km/hour (Casson, 1951). Starting from Abu Qir Cape and going reversely, along the coast, to the Arab's gulf, we find *El Alamein at* 124 km, Sidi abd El Rahman at 140 km and El Dabaa at 165 km, from the cape. Therefore, we conclude that the Argonauts, from Tritonis Lake, reached *in some way* the entire region of the Egyptian coast (from El Alamein to El Dabaa).


**3 Location of Tritonis Lake Entrance**

If we accept that the Tritonis Lake was located near the beach of Major Syrtis and the Argonauts could be able to detach the ship from the muddy shore, then they would continue normally their trip to Cyrenaica peninsula. But if it was impossible to
detach the ship, they would have to drag it to the coast and transport it through land in the eastern coast of Major Syrtis toward Benghazi in Cyrenaica peninsula. If we accept Tritonis' Lake location in Minor Syrtis, Argo should also have been directed to Benghazi, in Cyrenaica peninsula, in order to go to Crete. Of course, the Argonauts had no reason to head westward, from Major to Minor Syrtis. Their purpose is to approach Peloponnesus and not to move away from it. However, in the previous section, we have rejected all these cases and we have proved that Argo departed from the Egyptian coast, and
not from the Cyrenaica peninsula. The only option left is to accept Apollonius' description. The Argonauts decided to carry the ship away from Syrtis' shore to another bay, through the Libyan Desert, in order to continue their travel, since it was impossible to put the ship back into the sea, in the Major Syrtis Gulf.

Jason and the Argonauts received an important information (IV, 1312-1329) by 'three goddesses' (or native women?) and they saw a 'divine sign', when they were near the shore, two days after of their arrival to Major Syrtis. Specifically (IV,
1365-1379): '*from the sea to the land bounced a great horse ……And at once Peleus rejoiced and spoke among the throng of his comrades 'I deem that Poseidon's chariot has now been loosed by the hands of his dear wife'*. Argonauts, following the Poseidon's horse traces, have to transport the ship and cross the sandy ground of desert in order to go to another gulf. Indeed, the Argonauts transported Argo, '*with their strength and valor',* through the Libyan Desert to Lake Tritonis (IV, 1380-1392 and 1564-1570). Their travel lasted 12 days and nights. We note that Apollonius indicates that the Argonauts' route was '*ανά*
*θίνας ερήμου'- above the desert dunes'*, which means '*at the north edge of the desert',* because '*proposition ana (ανά) + accusative'* suggests a direction from bottom to top. At this point, it is necessary to make a few important clarifications:

A) The translation of the ancient phrase (IV, 1365-1379) '*ἀλλά μιν ἀστεμφεῖτε βίῃ καὶ ἀτειρέσιν ὤμοις ὑψόθεν ἀνθέμενοι ψαμαθώδεος ἔνδοθι γαίης οἴσομεν'-* '*But with unshaken strength and untiring shoulders will we lift her up and bear her within this country of sandy wastes'* gives the impression to the reader, by using the verbs 'lift and bear', that the *Argonauts*
*placed the ship on their shoulders*. However, there is only one verb in the text and this is '*οίσομεν*= we will bring'. Also, the



word '*ανθέμενοι*', originates from the verb '*ανατίθεμαι*' (past participle), means 'having assigned to our selves', utilizing their physical strength and their hard shoulders, in order to bring her (the ship) on ('*υψόθεν*') the sandy country.

B) (IV, 1380-1392): '*ἤ̈ βίῃ, ἤ̈ ἀρετῇ Λιβύης ἀνὰ θῖνας ἐρήμους νῆα μεταχρονίην ὅσατ' ἔνδοθι νηὸς ἄγεσθε ἀνθεμένους ὤμοισι φέρειν*' – '*by your might and your valour over the desert sands of Libya raised high aloft on your shoulders the ship and all that ye brought therein*'. And '*αὐτὰρ ἐπιπρὸ τῆλε μάλ' ἀσπασίως Τριτωνίδος ὕδασι λίμνης ὡς φέρον, ὡς εἰσβάντες ἀπὸ στιβαρῶν θέσαν ὤμων*'.- '*How forward and how far they bore her gladly to the waters of the Tritonian lake! How they strode in and set her down from their stalwart shoulders!*' The interpreter uses the verb '*raised*', while the referred verb is '*άγεσθε*', from '*αγω-drive*'. 'The word '*high aloft*' does not exist in the ancient text. According to the Lidell-Scott Lexicon, the verb '*άγω*' in passive voice '*άγομαι*', has the following translations: I take for myself, I receive, I take with me, I take something in my hands, I undertake, I attempt. In this case, '*άγεσθε φέρειν*'= '*you are attempting to transfer*' (the ship and its content). We remind the reader that the interpretation of the word '*ανθέμενους*' is '*having assigned to our selves*', through the strength of their shoulders to attempt to bring her (the ship) on the desert sands of Libya. Consequently, we point out that the Argonauts transported Argo, not on their shoulders, but *through the strength of their bodies and their shoulders,* for 12 days and nights. Also, the word '*εισβάντες*' originates from the verb '*εισβαίνω*' meaning 'embarking (boarding) into the ship', according to Liddel-Scott Lexicon. The correct interpretation is: '*And yet [they proceeded] far ahead until they transported her (meaning the ship) and deposited her through their strong shoulders, with joy, in the waters of Lake Tritonis, and then embarked in the ship*'. The Argonauts transported Argo, using their shoulders and their physical strength, through the Libyan Desert to Tritonis Lake.

C) (IV, 1564-1570): '*νῆα μεταχρονίην ἐκομίσσαμεν ἐς τόδε λίμνης χεῦμα δι' ἠπείρου, βεβαρημένοι*'- ' *have borne our ship aloft on our shoulders to the waters of this lake over the mainland, grievously burdened*'. The verb '*εκομίσσαμεν*' is mistranslated as '*have borne our ship aloft on our shoulders*'. *T*his verb ('*κομίζω*') has its meaning of carry, transfer, serve or bring something to somewhere. It remains with the same meaning even today, in the Modern Greek. When 'I carry-κομίζω' something is absolutely different from 'I carry-κουβαλώ' on my shoulders something. It means that, even in Modern Greek, there are two different verbs, which in English unfortunately are translated with only one verb. This causes misunderstandings. Consequently, the phrase '*εκομίσσαμεν δι'ηπείρου*' (the ship), is translated as *'we brought through the mainland'*.

D) Pindar and Apollonius give the same description for Argo's travel in North Africa (Pindar, Pythia, 4, 25-27 and *Scholia In Pindarum, by an Anonymous (Ode P 4, scholion 46, line 4)*). It is important to point out that Pindar also writes about the ship's transport '*υπέρ των ερήμων νώτων της γης*' (*over the deserted back of the land or over the desolated back places of the land*). Argonauts crossed the *deserted back* which means the *one edge of the desert* and not in the middle of it. Logically, this 'back' must be the northernmost part of the desert because it is closer to Major Syrtis' coast. Also, the word '*υπέρ*' does not mean 'over the desert' but 'on the desert*'*. That is, they do not raise the boat *above* the sandy ground but carry it *on* the sandy ground. For comparison, we mention the relevant verses from *Orphica Argonautica*, 230-248, which describes the

launch of the ship 'Argo' into the sea. According to that text, the ship was '*ὑπέρ ψαμάθου*', which means not 'over or above the sand' but 'on the sand'. Moreover, Pindar writes in the second of the above texts: '*καὶ μὴ δυνάμενοι διεξελθεῖν διὰ τὸ εἶναι τεναγώδη.., βαστάσαντες τὴν ναῦν διεκόμισαν εἰς τὴν Τριτωνίδα λίμνην'*- '*and not being able to pass through due to the shoal land,... they took the ship on their shoulders and carried it'*. The mistranslation gives the impression that the Argonauts raised the ship on their shoulders in order to carry it in the desert. However, the word *shoulders* do not exist in the ancient text. It uses the words '*βαστάσαντες'* and '*διεκόμισαν'*. The verb '*βαστάζω'* means I hold, I hold in hands, I lift up, I

raise, I carry and I transfer. The verb '*διακομίζω'* means carry or transfer. Thus, the correct interpretation is that 'they were pulling the ship by *holding it in their hands and carried it'*.

From the above analysis, we conclude that the Argonauts managed *to cross the Libyan Desert* in 12 days and nights in order to reach Tritonis Lake, where they placed their ship in it. They passed through *the north edge of the desert*, in a smooth area. They must have been traveling southeast, bypassing the Cyrenaica peninsula in order to approach a new coast *nearest to*

*Greece.* Of course, the Libyan Desert was extended mainly south and east of Major Syrtis' Gulf and Tritonis Lake has to be close to *Triton River,* which is *Nile River*, according to Apollonius. Argonauts transported the ship, using their strong shoulders, on the sand and not over the sand. They were holding the ship with their hands, dragging or pulling it and *in a way* carried it. The technology and equipment to drag Argo on the ground was known to them. Such technology existed and used, when they wanted to pull the ship into the sea and from the sea into any available sandy smooth coast *(Orphica*

*Argonautica*, 240-278 and Apollonius *Argonautica* I, 360-395). In both ancient texts there are many *detailed technical information.* They were placing many wooden rollers below the ship's keel connected with strong ropes. The Argonauts themselves tied with ropes and with their physical strength were pulling the ship.

However, it is impossible, for any person to travel '*twelve days and nights', nonstop.* The ship's transportation had to be carried out in a rotation system of two Argonauts teams. The first team could carry the ship in which the second team would

be boarded and then vice versa. Maybe this is suggested in the phrase that they transferred '*the ship and all that ye brought therein'*. Also, we assume that Apollonius provides the information of 12 days and nights route, as a *distance measure*. Indeed, Herodotus frequently describes the distance between two locations in '*a day's duration'* (e.g. in II, 4, 29, in IV, 181-184). Thus, he mentions that the distance between Siwa Oasis and Awjilah Oasis (which are in the Libyan Desert) was covered in 10 days (IV, 182). This information was confirmed by the German explorer Hornemann (1772–1801) who

covered the distance in 9 days, although caravans normally take 13 days ([https://en.wikipedia.org/wiki/Awjila](https://en.wikipedia.org/wiki/Awjila)). This means that a person can cover the distance Siwa-Awjilah of 395 (~400) km, in a straight line on the Google earth map, with the speed of about 39.5 (~40) km per day and night (or 1.7~2 km/hour). This is a minimal distance. Due to possible geomorphological obstacles of the desert, the 'real' distance should be bigger.

The average speed of a walking person is about 4 km/hour (Browning et al. 2006). In our analysis, we accept as *working*

*hypothesis* a decreased speed of *2 km/hour* (48 km per 24 hours) taking into account the carrying of the ship. Consequently, the Argonauts would have traveled *at least 576 km* in 12 days and nights, nonstop. This distance is comparable with the

distance between the coast of Major Syrtis and Siwa Oasis (Lake Maraqi), in a straight line on Google earth map, which is equal to 594 km. We are forced to accept that the Argonauts arrived to *the ancient Tritonis Lake,* which was located in the *cluster of lakes* in Siwa Oasis.

According to Apollonius, the Argonauts *do not travel randomly* through the north edge of the desert. They follow the 'footprints of Poseidon's horse' which were on the ground. It is known that Poseidon is the God of water and his horse hits the ground and gushes water. Thus, we conclude that these 'traces' must be *the water route* existing in the 13th century BC, in the northern part of the Libyan Desert, in the form of shallow lakes or rivers, one after the other. These waters are the remnants of a previous palaeoriver system thousands of years ago. Indeed, in accordance with satellite data provided by

Ghoneim and El-Baz (2020), below the desert's terrain rich groundwater deposits exist (Fig.2 (left)). The authors write: 'in the past, this region underwent drastic climatic changes alternated with dry to wet phases' and 'During wet phases, when the rain was plentiful over a prolonged time period, the surface was veined by rivers and dotted by large lakes'. Their findings are fully compatible with the references of other authors for the wet phase of 13th century BC (Fig. 2 (right)), during which the Argonautic campaign taking place (e.g. Gasse, 2000). Even today, at the northward of the Libyan Desert, there are such

water residues in the form of two wadis (Wadi el Farigh and the Wadi el Hammim), one sabkhat (Sabkhat Shunayn) and many lakes in Jaghbub (Tobruk) Oasis (*https://en.wikipedia.org/wiki/List_of_wadis_of_Libya#/media/File:Un-libya.png*).

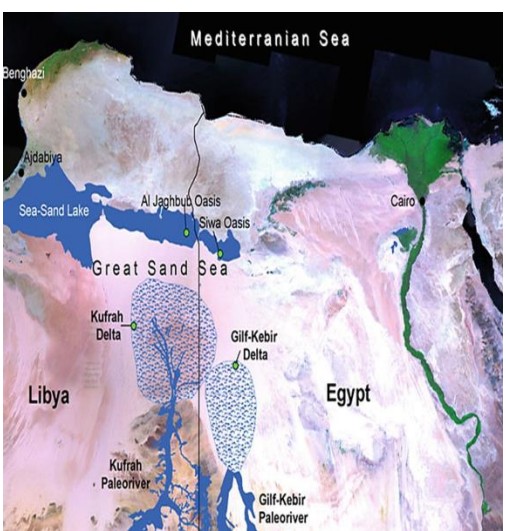
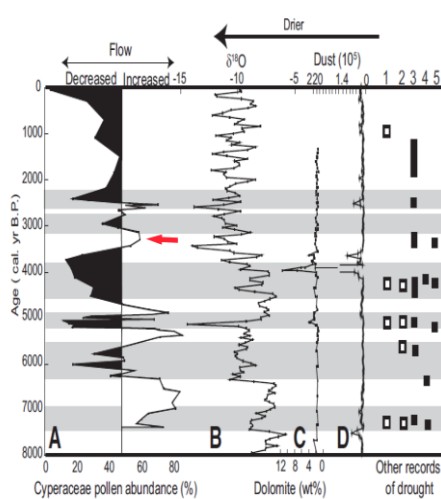

*Figure 2 (Left) The system of paleorivers by Ghoneim and El-Baz (2020) is shown. It can be seen that the whole region between Major Syrtis and the Siwa Oasis was covered with underground water. (Right) The increased and decreased flow of*

*water in the lakes of Eastern Africa as a function of time by Bernhardt et al. (2012).The red arrow shows the increased flow of waters during the period of the Argonautic Campaign, in 1300 BC (3300 B.P.).*

We used Google Earth software in order to define Argo's path in the northern edge of the Libyan Desert, passing it, inside of these sabkhas, wadis and oasis (Fig. 3). This route has a mean altitude of 36m and a mean slope 0.3-0.4%. The distance of 665 km starts from Sabkhat Ghuzayyil (considering it as the old coast of Major Syrtis) and ends to the west edge of Siwa

Oasis. Argo crossed the system of many successive lakes of Jaghbub Oasis which are very close to each other. According to Murray (1952), the underground water is abundant in the region of Jaghbub and Siwa Oases and between them there are 45 km$^2$ of lagoons, some of them are 3m deep. This environment, before thousands of years was wetter and consequently would help the transport of Argo through them. It's also crucial to note that in today's conditions, this route contains numerous little water pockets along around 70% of its length. We can deduce that in the circumstances in the 13$^{th}$ BC, the same percentage

of 70% would have significantly more water allowing Argo's sailing. This percentage increases to more than 70%, due to the possibility that the waters were also covering additional areas.

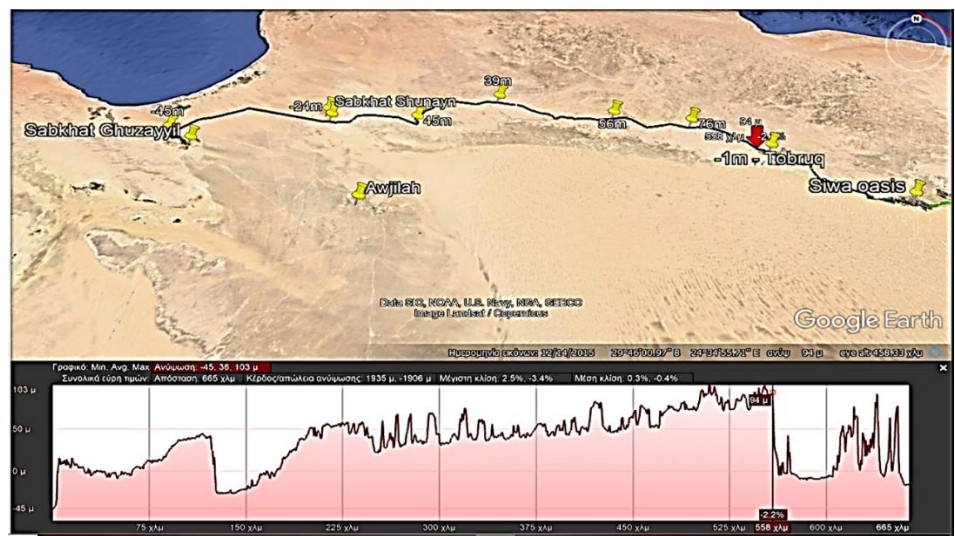

Figure 3 The 665 km route from Sabkha Ghuzayyil (Major Syrtis Gulf) to Siwa Oasis, crossing today's wadis, sabkha etc,
with an average elevation of 36m and a mean slope of 0.3-0.4%. In the last part of this route are two oasis; the Jaghbub Oasis (Tobruk) and Siwa Oasis. We note that the maximum altitude remain below or equal to 75 m, except for a distance of ~100km, which the altitude increases to 103 m. However, in the entire region there are small hills interrupted by flat patches of terrain. In such a case, the route could bypass a high hill passing right next to it, at a lower altitude. © Google Earth

This proposed route of 665 km must have been covered within 12 days and nights. This is equivalent with an *average speed*
of 2.3 km/hour which is slightly bigger than our estimated speed of 2 km/hour. However, it is accepted because is the speed of a combined way of transportation, meaning (a) navigation (Argo was sailed slowly and carefully within the shallow rivers/lakes), (b) walking (Between the successive shallow rivers/lakes, the Argonauts transported the ship using the known

technology mentioned earlier in the text) and (c) both together (The Argonauts had to pull the ship which was within shallow water because its keel almost would touch the muddy bottom). The time of the effort was increasing, when Argo was

occasionally outside the aquatic environment, because the speed of navigation was higher than the speed of dragging the ship, with ropes.

The methodology, for the above mentioned case, was known to them, as described by *Orphica Argonautica* (1097-1106):'*They jumped out of the ship's sideways walls in to the very shallow sea, coming up to tenacious mud in the sea. Quickly they tied the ship with the special designed strong ropes, by connecting with a long rope the ending part of the stern.*

*Agaeus and Argus threw that long rope to the heroes in order to catch it. Then, they were dragging the ship, since they themselves being on the shore and they were moving fast'.*

Siwa Oasis is a depression with an average depth of 18 m below sea level (9-28 km). From east to west has a length of 82 km and covers an area of 1200 km$^2$. They exist there many mountains and hills as well as many *salty lakes* and many *fresh water springs*. We accept that the four largest lakes communicated with each other, in the wet conditions of the 13th century

BC, due to their proximity. Thus, they all together formed one large single lake (Tritonis Lake). A proof that our calculations are correct, originates from Apollonius' text (IV, 1312-1317 and 1325-1329): After the Argonauts put the ship in the lake, *they were looking for a water source* because they were exhausted and thirsty. How is this behavior justified when there is a whole lake in front of them? The only explanation is that the lakes of Siwa Oasis offer salty water, which is non-drinkable. There are two more Apollonius' descriptions which betray an oasis' environment: A) the thirsty Hercules appeared '*like*

*anyone else who crosses this area on foot'*. It is known that those who cross a desert suffer from thirst. B) Argonauts tried to find Hercules, but '*his traces on the sand had vanished due to the wind which transported sand and covered them'*. Sandstorms are a common phenomenon in the deserts which result in the movement of sand.

## 4 Siwa Oasis: Tritonis Lake and the Garden of Hesperides

According to Apollonius, the Argonauts arrived in Tritonis Lake, where Hercules had arrived also one day earlier (IV, 1400-

1448) and he had already taken the apples of the Hesperides (11th labour). The nymphs were frightened by the sudden appearance of the Argonauts and they transformed into '*dust and soil'*. However, after the intervention of Orpheus, the three Hesperides became *hydrophilic trees (a poplar, an elm and a willow)* before taking again their divine form. We believe that this is a poetically description of an oasis (with grass, saplings, flowering shoots and hydrophilic trees) between the 'dust and soil' of the desert. Apollonius clarifies that this '*ιερόν πέδον- sacred plain'* is the *'Άτλαντος χώρος-Atlas' farm'*. It is known

that after the Titanomachy, Zeus forced Atlas to leave its farm and placed him at the top of Atlas Mountain, in north-west Africa. There, he is holding eternally on his shoulders the sky (Pherecydes, *Fragmenta,* 33; Hesiodus, *Theogony,* 517-518). Hera (Zeus' wife) planted the apple trees, which was a wedding gift by Gaia, in the '*garden of the gods'* which was located in the previous '*Atlas' farm'*. However, Atlas' daughters were keeping stealing the apples and for that reason Hera entrusted



their care to the nymphs Hesperides and a dragon-Ladon. Consequently, *the garden of Hesperides* or the *garden of the gods* or the *'older farm of Atlas'* characterized as the '*sacred plain*' was near to Tritonis Lake, in Siwa Oasis.

The existing sacredness of the location is also proved by the location of the ancient *oracle of Ammon* near to Al-Zaytun place, on the hill named *Aghurmi,* in a plateau, with a height of 30m (http://www.siwa-oasis.it/aghurmi.html). In front of it, in a NE direction, there are the 'famous gardens', near *Al-Zaytun Lake.* The today's temple of Ammon-Zeus was built during the 26th Dynasty by pharaoh Amasis (570-526 BC), upon an earlier sanctuary. The older sanctuary was founded by Danaus,

twin brother of Aegyptus. These are the children of Belus, King of Libya, which was the son of Poseidon and Libya (Diodorus of Sicily XVII, 50, 2, 2). We note that Argonauts arrived in this place, following the *'footprints'* of Poseidon's horse (the remnants of an older comfortable *water route*) whose grandson, Danaus, founded the temple. This myth 'transfer', in some a way, the information of the connection-the known route between the Major Syrtis Gulf, in Libyan coast and the 'sacred plain' of Siwa Oasis.

It is well known that ancient societies linked the rebirth of nature with the matching zodiac sign each time the Sun entered the sky on the Spring Equinox (Stageiritıs, book 6, chapter 7). Thus, the symbol of the solar deity, Ammon-Zeus (*κερασφόρος-kerasphoros,* 'horned god'), was the ram, because after 1800 BC (to 1 AD), the Spring Equinox is taken place in the zodiac sign of Aries (ram). According to Eratosthenes (Catasterismi, 1, 19), this zodiac represents the '*golden ram'* that transported Frixos and Elli, King Athamas' children, from Orchomenus to Aia/Colchis. However, the Argonauts have

with them, the *Golden Fleece of this ram*, according to Greek mythology, when they arrived in Siwa Oasis. This 'coincidence' and the 'unlikely arrival of a ship' in an oasis it cannot have gone unnoticed by the ancient priesthood. Quintus Curtius (*History of Alexander, 4. 23-25*[1]) and Diodorus Siculus (*Bibliotheca Historica,* XVII, 50, 6, 1–7, 2[2]) recorded a very ancient tradition, in connection with the very original form of a religious activity of the Ammon-Zeus' oracle, which remained even in their own time, in the first century BC. A litany of a *'golden ship'* was taking place. It carried by 80 priests.

A sacred stone or xoanon (as the image of the God) of conical shape, defined as *'navel'*, existed inside the ship which was

---

[1]*What is worshipped as the god does not have the same form that artificers have commonly given to the deities; its appearance is very like that of a navel fastened in a mass of emeralds and other gems. When an oracle is sought, the priests carry this in a golden boat with many silver cups hanging from both sides of the boat; matrons and maidens follow, singing in the native manner a kind of rude song, by which they believe Zeus-Ammon is propitiated and led to give a trustworthy response.*

[2]*The image of the god (xoanon) is encrusted with emeralds and other precious stones, and answers those who consult the oracle in a quite peculiar fashion. It is carried about upon a golden boat by eighty priests, and these, with the god on their shoulders, go without their own volition wherever the god directs their path. A multitude of girls and women follow them singing hymns as they go and praising the god in a traditional hymn.*



ritually transported. This strange object *was represented* the kerasphoros (who bears ram horns) Ammon-Zeus. Women followed and chanted, as the ship was being transferred by the priests. Could this unusual ritual illustrate the Argo's arrival in the oasis? The 80 priests could be represented the number of 50 rowers (*Orphica Argonautica* V.278-306), and other auxiliary personnel of Argo. Women could be represented Medea and her followers. The Argonauts carried the ship with the

strength of their shoulders, wandering in the desert, transferring the *Golden Fleece of the ram (*and *not the ram itself), on the ship.* Exactly, as '*the unusual image of the god'* and *not the* '*horned god' himself (the usual form of the God)* is transferred by the priests, on the *golden ship.*

A necessary prerequisite, for the above hypothesis, is the existence of the temple, when the Argonauts arrived there, in 13th century BC. We note that Ammon rose to the position of patron deity of Thebes, from the 11th Dynasty (21st century BC).

During the reign of Ahmose I (1550-1525 BC), *Ammon* gained national significance, as a result of the merging of the two Egyptian religious centers of Hermopolis and Heliopolis, and renames *Ammon-Ra*. Between the 16th and 11th centuries BC, this deity had the title of 'King of Gods'. Pindar writes (Pythian 4) that Ammon had become synonymous with Zeus since both were regarded as "King of the Gods" in their own pantheons. Indeed, there is a myth connected chronologically the oracle in Siwa Oasis with the oracle of Zeus, in Dodona (Greece). Two priestesses (or two doves) flew away,

simultaneously, from the exited Egyptian Thebes' Temple (Herodotus, 2.54-58). One of them reached to the oracle of Gaia, in Dodona, and founded the oracle of Zeus, in the same place. The other came in Siwa Oasis, where she established the Ammon-Ra Temple, transforming the pre-existing Ammon oracle. However, the earliest mention of the oracle of Zeus in Dodona is in Homeric Iliad (Il.16, 233-235) and Odyssey (Od.14, 327-328). Thus, in the 13th -12th century BC, during Trojan War, the oracle of Dodona (and the oracle of Ammon-Ra too) already existed.

Even, according to the Egyptian mythology, the oracle of Ammon existed in Siwa Oasis, in very old times. Diodorus of Sicily (I, 13, 2, 4 and III, 68-74)) writes that Dionysus, son of Ammon and the nymph Amalthea, army abbot, reached Siwa Oasis, driven by a ram and built the oracle, in honor of his father, during the Titanomachy (Titans' War against Gods) epoch. The connection between Titanomachy and the oracle is hidden in the orientation of its walls. There is a suitable arrangement so that a ray of sunlight illuminates exactly the center of the temple *during the winter solstice* ([http://www.siwa-](http://www.siwa-oasis.it/aghurmi.html)

[oasis.it/aghurmi.html](http://www.siwa-oasis.it/aghurmi.html)), which was the zodiac sign of Capricorns, after 2200 BC (to 100 AD). It was marked, symbolically, the end of the Titanomachy with victory of the gods, representing the *god Pan* who saved Zeus from the Typhoon, according to Eratosthenes (Catasterismi, 1, 27). It is noticeable that, Zeus reached Atlas' farm (in Siwa Oasis), after his victory, and founded the 'garden of the gods' in it.

Siwa Oasis is now connected to Marsa Matruh's coast (the westernmost point of the coast, from where the Argonauts will

departure, on their way to Crete) by an avenue as it is located 300 km from the Egyptian coast. The question which arises is how the Argonauts from Siwa Oasis arrived in the Mediterranean Sea, by sailing?



**5 Tritonis Lake and Qattara Depression**


A sizable place of Egypt between the coast and Siwa Oasis is occupied by the Qattara Depression. With a depth of 133-145 meters below sea level, this depression is the largest in Africa. It covers over 20000 km$^2$ between the longitudes of 26°20' and 29°02' east and the latitudes of 28°35' and 30°25' north. There are several sand dunes, salt pans, salt marshes, and large lake bottoms inside it. The marshes occupy approximately 300 km$^2$ (https://en.wikipedia.org/wiki/Qattara_Depression).

About a quarter of this depression (4901 km$^2$) is occupied by dry lakes composed of hard crust and sticky mud, and occasionally filled with water. The Qattara Depression was once a vast lake thousands of years ago. It lost a significant amount of water due to climate change, yet it preserved a significant amount below its dry level (Albritton et al, 1990, Aref et al, 2001).

According to Aref et al (2002), 'over large areas of the floor of the depression, the bedrock is covered by younger deposits

including wind-blown sand, sabkhas, and Quaternary evaporates, sediments. During the wet periods of the Quaternary, groundwater and rainfall resulted in the rise of the water table, which fed the inland *sabkhas* (Fig. 4). During the dry periods of the Quaternary, evaporation processes predominated, where salt weathering disintegrated the bedrock of the *Moghra or Dabaa Formations*. Sandy and clayey layers of the Lower Miocene *Moghra Formation* form its bottom and surroundings, where the elevation ranges from 50 to 80 m below sea level. In some areas, the Moghra sediments occur as small plateaus

and dissected hills within the sabkhas. The Upper Eocene-Oligocene *Dabaa Formation* underlies the south-west part of the depression, including all areas 100 m below sea level. It consists of black shales and contains abundant gypsum veins and shark teeth and remains. The northern *Diffa Plateau* separates the Qattara Depression from the Mediterranean frontal plain. The northern steep escarpment is associated with the Middle Miocene calcareous sediments of the Marmarica Formation, with a thickness of a few meters at the rim of the depression, increasing to several hundred meters at the coast, where

Pliocene carbonate are exposed'.

Based on the above description, we estimate that the *Dabaa formation* and the *sabkhas* cover an area of about 6000 km$^2$ extended from the southern to the northern part of the Qattara Depression (Fig. 4). The increased of the water flows in the 13th century BC (see in Fig. 2 (right)), forces us, to accept the existence of satisfactory volume of water in the Dabaa formations and in the sabkhas which allowed Argo to sail through them. In Fig. 5, this route of Argo, from South to North,

inside the Qattara Depression is shown. The proposed course is based on the topographic and geologic map of Fig.4.

The Siwa Oasis is expanded in 50-70 km south-west of the Qattara Depression. The existence of underground waters, below the surface of this, between them, region is depicted in Fig. 2 (left). Also, in this area, there are some small oases. The Siwa Oasis appears to be connected to the Qattara Depression by a palaeoriver marked with a letter A in Fig. 6. Its northern and southern altitudes of 'A- region', which has an average altitude of about 8m below sea level, are around 80 and 50 meters,

respectively. As a result, we deduce that the water level in the 13th century BC permitted Argo to sail from Siwa Oasis to

Qattara Depression (Fig. 6). Three dunes with a height of about 28 m can be seen in this proposed route. It is very likely that they formed much later, as the desert expanded north. In any case, the Argonauts would have simply sailed around them. Consequently, whereas Tritonis Lake was located in the Siwa Oasis, it was spreading eastwards, into the Qattara Depression. Probably, this information is contained in the phrase '*λίμνης χεύμα- water stream of the lake'* (Apollonius Argonautica, IV,

1569) characterized this extension, when the Argonauts were inside the Qattara Depression, according to our hypothesis. The word *'χεύμα'* derives from the verb *'χέω'* which means *'pour'*. As a result, this distinctive phrase depicts a water continuum between Siwa Oasis, which was once part of Tritonis Lake, and the rest of Tritonis Lake, which is now the Qattara Depression.

Arriving from Siwa Oasis, the Argonauts were moving northward *('helped by the south wind',* according to Apollonius),

through Qatarra Depression in order to approach the Mediterranean Sea, which is north of Qatarra Depression but not directly connected to it. This Argo's route is described with details by Apollonius (IV, 1537-1547): *'But when they had gone aboard, as the south wind blew over the sea, and they were searching for a passage to go forth from the Tritonian lake, for long they had no any idea, but all the day were sail on aimless'.* The ancient text uses the word '*πέλαγος-sea'* meaning that this area was of enormous dimensions. Additionally, the word '*δηναιόν χρόνον-long time'* means that the Argonauts were

sailing possibly for days, in order to find the exit from the lake. Because there is no visible waterway leading from the Qattara Depression to the Mediterranean Sea, the Argonauts asked the local gods for assistance. Then, 'God Triton' himself manifested as King of Cyrene, Eurypylus, son of God Poseidon, and he enquired of them whether they were seeking a route between the lake and the sea (IV, 1554–1561): *'But if ye are searching for a passage through this sea,… I will declare it. For my father Poseidon has made me to be well versed in this sea…'.* The Greek words '*ἄλς and πόντος'* translated correct

as *'sea'* are indicating also the large size of this area. All these characterizations fit well with the large area of Qattara Depression.

According to Apollonius (IV, 1571-1576): '*Triton-'Eurypylus' stretched out his hand and showed afar the sea and the lake's deep mouth, and then addressed them'.* The large area and the great depth of the lake are expressed with the ancient Greek words '*πόντος-sea'* and '*ἀγχιβαθές- is deep up to the shore'*, respectively. We accept that the Argonauts sailed from South to

North and reached near to Moghra Lake (Fig. 5) which is 4 km$^2$ lake containing brackish water, marshes and a swamp, with altitude some 38 m below sea level. North of it, is a cliff which gives its name to the Moghra Formation, a thick layer of clastic sedimentary rocks with some minor carbonate interbeds. According to Khan et al (2014), Moghra Lake is a remnant of a larger palaeolake. They also demonstrated the existence of *the mouth of a palaeo-river* in this region with direction from east to west (Fig.10 in their publication).


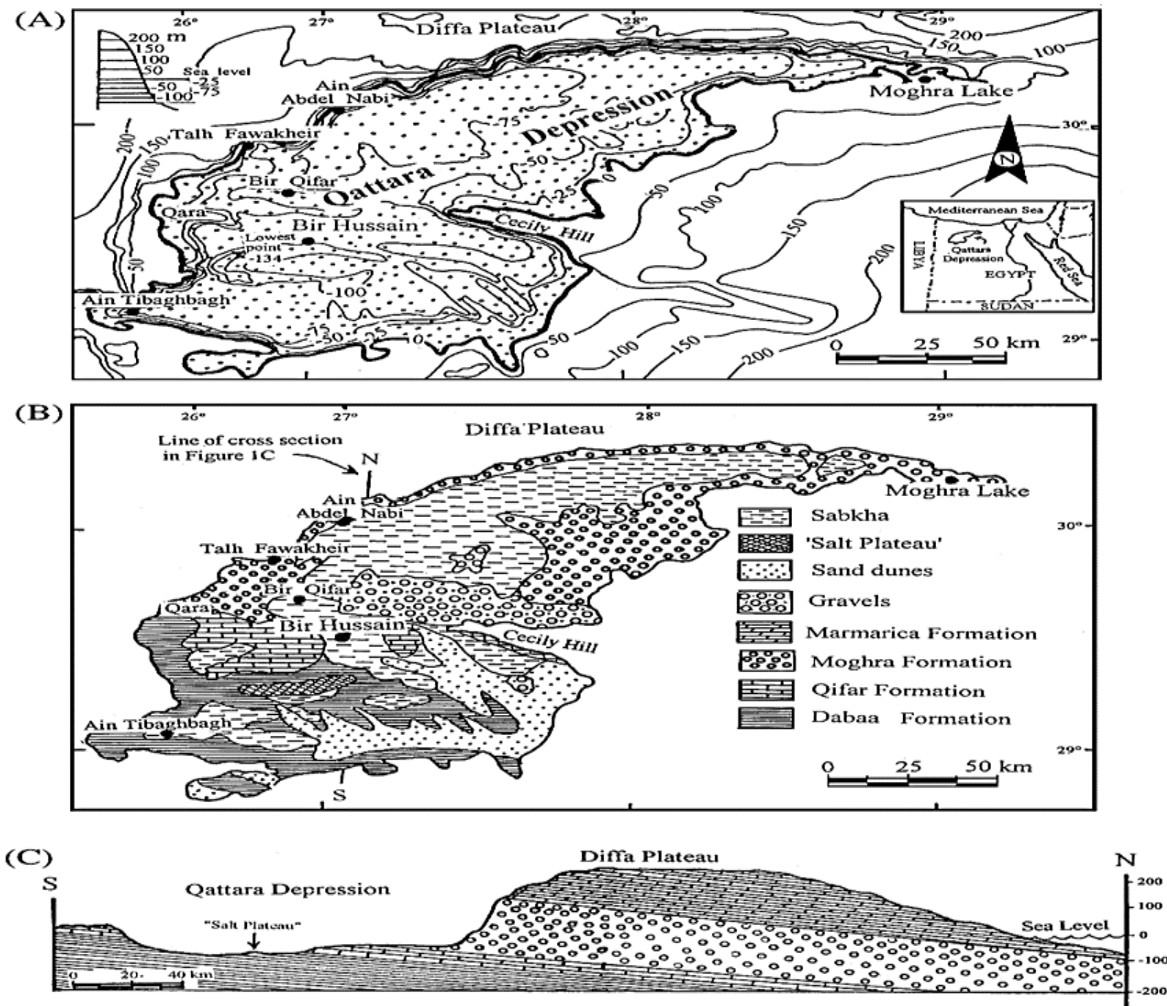

*Figure 4 Topographic and geologic map of Qattara Depression after Aref et al (2002) and related references to it. See for a detailed description in the text.*

And the 'God Triton' continued: *'That is the outlet to the sea, where the deep water lies unmoved and dark; on each side roll white breakers  with shining crests; and the way between them for your passage out is narrow'.* The ancient Greek word *'ρηγμίνες-breakers'* depicts an environment comparable to that found on a steep and vertically carved' coast, where the sea waves crash. In this case, the same word describes the high, steep and rocky (northern) edge of the Qattara Depression/Tritonis Lake (characterizing as 'sea' due to its large size), which consists of 180 m above sea level hills. These white features on the hills are caused by salt depositions, and they still be seen across the Qattara Depression, today. The lake's *'διήλυσις-outlet'* to the Mediterranean Sea is located in the middle of these 'high and rocky exposures'. This ancient word is produced by the verb *'διελαύνω'* which has two meanings; *'penetrate-διαπερνώ'* and *'pierce-διατρυπώ'*. This means



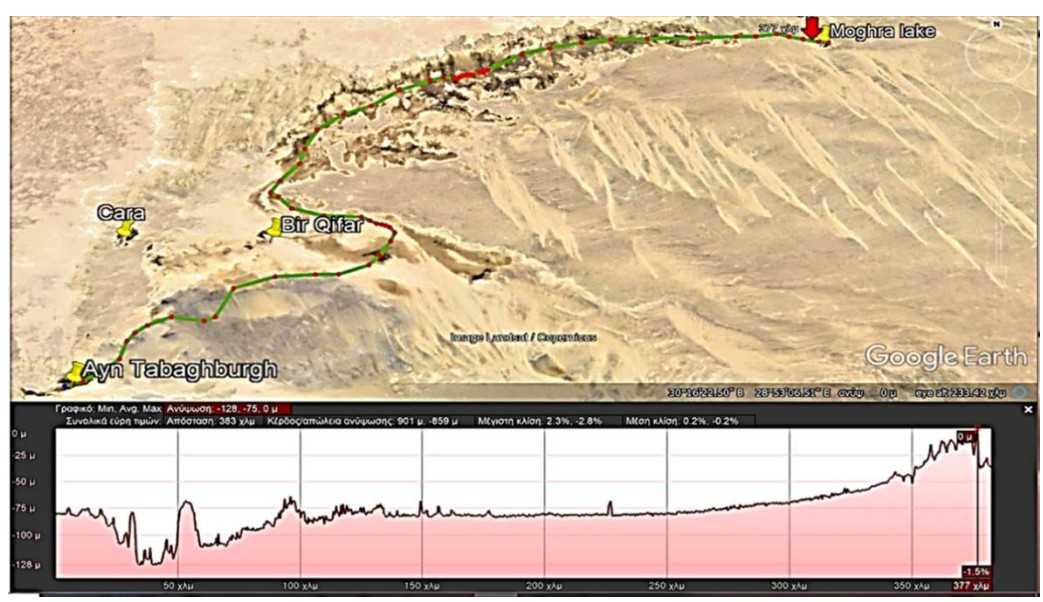

*Figure 5 Argo's route inside Qattara Depression, according to our analysis. From Siwa Oasis, following a palaeoriver, the*
*ship entered in Qattara Depression (Ayn Tabaghburgh). Argonauts sailed from South to North through the 'today's' Dabaa*
*formations and the Sabkhas. The end of this route is near to Moghra Lake. The elevation profile is given at the bottom part.*
*All this area of the route remains below sea level. © Google Earth*

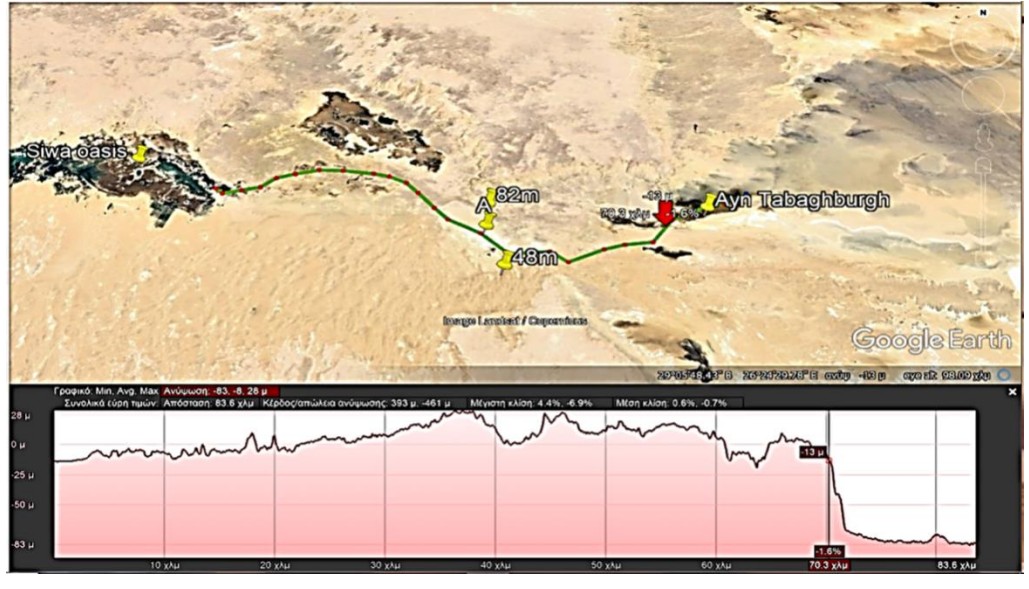

*Figure 6 The proposed route of the Argonauts from Siwa Oasis to Qattara Depression is shown. It seems that there is a*
*palaeoriver (marked with 'A') which connects the Siwa Oasis depression with Qattara Depression. The 'entrance' to Qattara Depression is characterized by the sharp decrease of the elevation profile at values corresponding to altitudes below sea level. The elevations northern and southern of this route are about 80 and 50 meters respectively. © Google Earth*

that Argo sails along a *narrow* waterway which could be either a *physical water lane* or an *artificial canal* passing through the craggy hills**.** It is known that before the 13th century BC, the Egyptians had the technology to create man-made lakes and
canals (e.g. Herodotus II, 99-101 and II, 149).

The Argonauts boarded the ship and began rowing. Triton himself, with his 'divine face', guided the Argo through this narrow passage and took her out to the sea (IV, 1609-1610 and IV, 1617-1618). It seems that sailing through this *narrow passage* was extremely difficult, requiring "divine aid" to be successful. This Apollonius' description probably implies that the local population (e.g. the King of Cyrene) was aware of this passage connecting Tritonis Lake to the Mediterranean Sea
and so they gave the 'relevant exit instructions' to the Argonauts.

**6 Location of the Narrow Passage**

After thousands of years, the position of the narrow water passage, which obviously does not exist today, is difficult to detect due to geomorphologic changes that have occurred in the particular area. Nevertheless we tried to approach it creating a Digital Terrain Model and using geo-information techniques to determine *the thalwegs* that lead from the Moghra Lake
district to the Mediterranean Sea. Because the 13th century BC is before the 'dry period' of the 'dark ages' (see in introduction), our working hypothesis is that *then* there was a flow of water in most of these thalwegs. The 'narrow passage' should coincide with some of them.

Our steps were the following: A) We found and used digital elevation data from i) NASA and in particular satellite data from Advanced Spaceborne Thermal Emission and Reflection Radiometer (ASTER). The latter obtains high-resolution (15 to 90
m$^2$/pixel) images of the Earth in 14 different wavelengths of the electromagnetic spectrum, ranging from visible to thermal infrared light (https://asterweb.jpl.nasa.gov/data.asp), ii) Topographic maps made by the American Army in a scale of 1:250000 and with contour lines of the altitude with 20 m spacing (https://maps.lib.utexas.edu/maps/ams/north_africa/) and iii) GIS data from DIVA's site (https://www.diva-gis.org/gdata). B) We extracted the digital elevation model (DEM) of the area of study (Fig. 7 (top)) working on QGIS software (version 3.6 Noosa). C) In order to define the hydrographic network,
we utilized the TauDEM (Terrain Analysis Using Digital Elevation Models), as a tool, in QGIS's environment. With this, we defined *the digital lines of thalwegs,* in other words the lowest bathymetrically, sub-regions, in which cliffs opposite to each other, meet. These sub-regions are named *'water lines of flow'* and illustrate the natural waters' direction. The digital hydrographic network of our study area in which the natural waters' directions can be recognized is presented in Fig. 7 (middle). D) From the above hydrographic network we have chosen 'the waters' lines of flow' which connect Moghra Lake

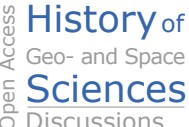

region with Mediterranean Sea and *probably had water in the 13th century BC*. Thus, five possible scenarios of these 'waters' lines of flow' (E0, E1, E2, E3 and E4) were produced and they are shown in Fig. 7 (bottom). However, we exclude the case E0 from our analysis, because the Major Syrtis was the starting point of the Argonauts' adventure to the desert. The other four scenarios are presented below:

- Case E1 (Exit towards Tobruk): The E1 path is replete of oasis, lakes, and water concentrations; *however the coast
is not accessible by this route*. The Argonauts would have to direct themselves for 200 kilometers towards higher regions, commencing at an altitude of 50 meters and climbing to a maximum altitude of 190 meters in these highlands with the river flow always opposing their route.

- Case E2 (Exit towards El Alamein): This 180 km route has, mainly, gentle and downhill slopes. There is only an 18 km length of increasing altitude from 50 to 100 m, but with low slope. After that, up to the coast, the sailing of the
ship is done according to the direction of the water flow.

- Case E3 (Exit towards Alexandria): This 235 km route has mild and downhill gradients. It also includes the 18-km stretch of increasing altitude from 50 to 100 m (see E2 case). We assume that it is improbable Argonauts chose this longer inland route when they could reach the sea very soon via the E2 route.

- Case E4 (Exit towards Wadi El Natrun): This water route has exceptional topographic difficulties, mainly due to the
very big slopes of the terrain.

Therefore, exit E2 represents the Argo's route with the highest probability (see also in Fig. 7 (middle)). In Fig. 8, the altitude variation profile of E2 route is displayed, with a mean altitude of 51 m and an average slope of 0.5%. All potential depressions (deirs) presented along this route are marked by blue color on the topographic map (Fig. 9). Because they are large enough, could have had significant amounts of water during the 13th century BC, just before the
'dry period'. It is very likely that some of them represent palaeo-lake locations.

According to the above results, we describe the following Argo's path: A water route connects Moghra Lake to three successive depressions (Fig. 9), which are spaced apart by 2-3 kilometers. Due to the low slope, the Argonauts were able to sail up to 50 m altitude on this particular water path (3rd depression named El Faiyada in Fig.9). These depressions are the remains of a palaeo-river which it is identical to one of those characterized as 'present day water-flow direction' presented
by Khan et al (2014) in their Fig. 3 (see in Fig. 10). This water route bends west and leads to an 18km-long gorge (marked by the arrows in Fig. 9). The altitude at the beginning of this gorge is around 50 meters, while it is about 100 meters at the end. This short path (the gorge) passes between hills which are 120 m high (see iso-countour lines in Fig. 9). Also, this narrow passage is shown in Fig. 8 (right), in Diffa Plateau which is the highland range in the upper limit of Qattara Depression.

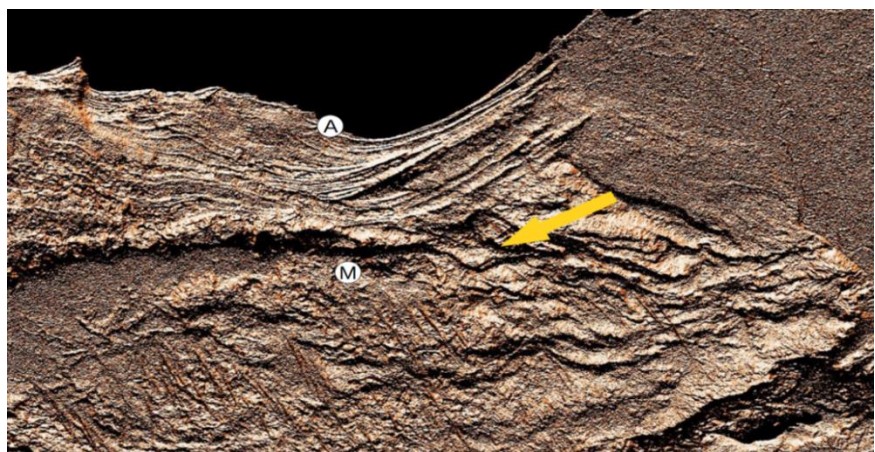


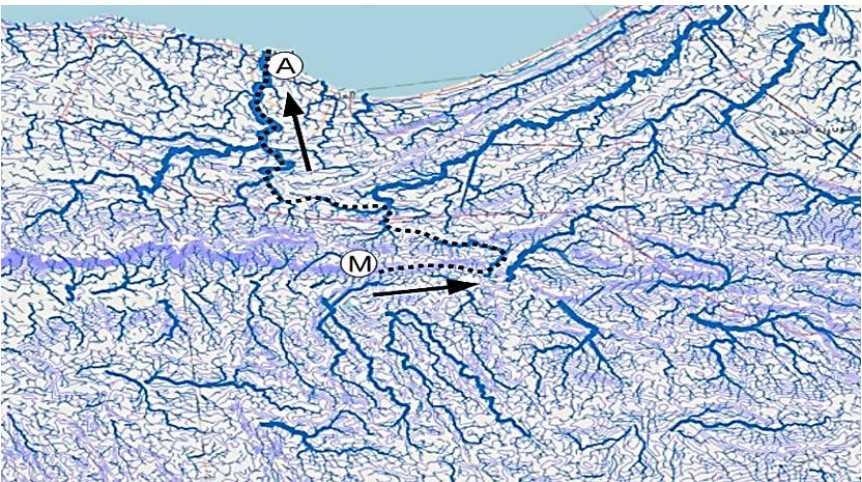

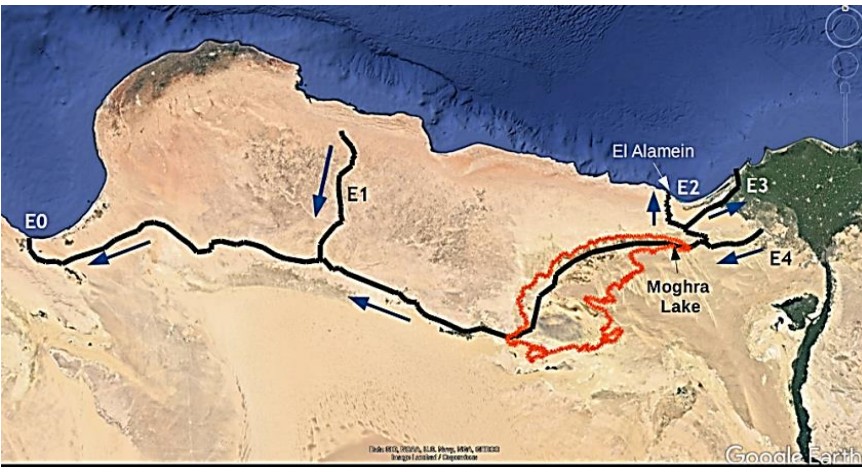

*Figure 7 (top) Part of the DEM produced in the Qattara Depression area (© NASA Advanced Spaceborne Thermal Emission Radiometer & © DIVA-GIS ). The yellow arrow indicates the narrow passage (middle) The hydrographic network of the*

*wider region of Qattara Depression is shown (bottom) The five possible scenarios - paths (E0 TO E4) which resulted from the present study is presented. The final route which selected (E2) is shown in detail on the hydrographic map. The A and M on the figures show the position of El Alamein and the Moghra Lake respectively (© Google Earth). The arrows indicate the direction of water flow ((© NASA Advanced Spaceborne Thermal Emission Radiometer & © DIVA-GIS).*

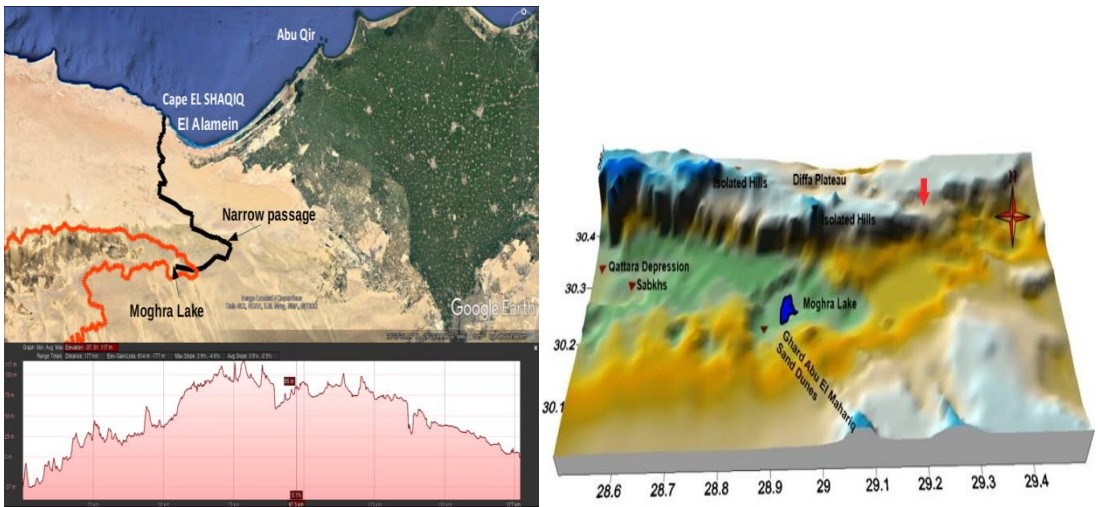

*Figure 8 (© Google Earth)  (left) The route of the Argo (black line) from Qattara Depression (Moghra Lake) through the narrow passage (black arrow) to the Diffa Plateau and then the downing course to the coast, near to El Alamein. The limit of the Qattara Depression is marked with a red border line. The elevation profile (mean value 51m) and the mean slope (0.5%) of this route are given to the bottom diagram. It is noted that this route is passing continuously through large depressions-palaeolakes (see in Fig. 9). Right) The 3-D surface, which show a part of the Qattara Depression, Moghra Lake and Diffa Plateau (Yousf et al (2018)). The arrow which was added by us shows the narrow passage.*

Additionally, this gorge coincides with an old fault as it is shown in Khan et al (2014)'s Fig. 3 (near to El-Hagif Fm). Even if there was no water in this gorge in the 13[th] century BC, Egyptians possessed the technology to build a channel, taking advantage of the existence of the fault, in order to create a water route from the old Tritonis Lake to the Diffa Plateau and eventually to the Mediterranean Sea. At the end of this ravine (narrow water passage), two large palaeo-lakes exist in the

Diffa Plateau (El Seneb and El Ragil, see Fig. 9), probably relics of an older lake, which still existed in the 13[th] century BC. The Argonauts were able to sail against the opposite water flow, due to the low slope. From Diffa Plateau, the altitudes are steadily reduced up to the coast and there are numerous depressions along the proposed route, which betray the functioning of palaeo-lakes (Fig. 9).  This water route, E2, reaches to the coast at Sidi Abd El Rahman, located 20 km west of El Alamein.



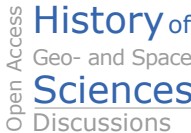

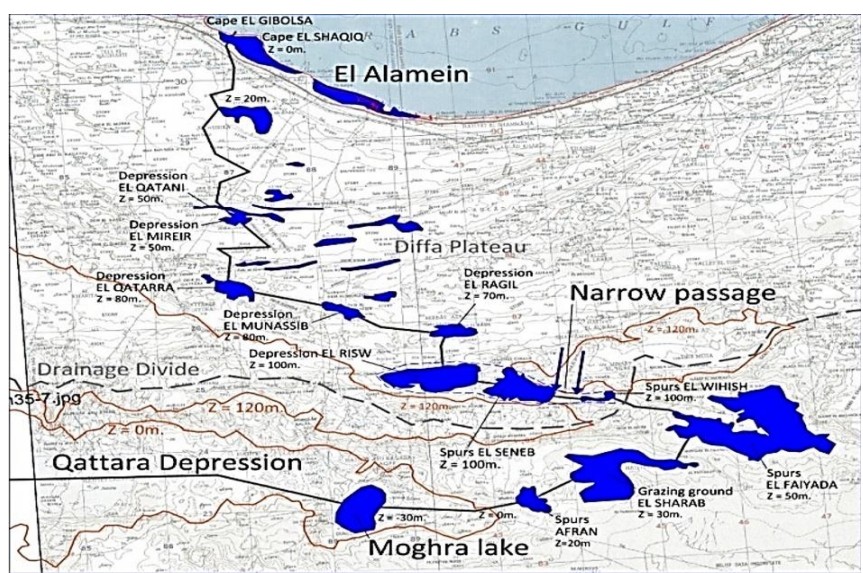

*Figure 9 Topographic map of the area made by the US Army, at a scale of 1: 250000 and isometric dimension of contour lines of 20m (https://maps.lib.utexas.edu/maps/ams/north_africa/). With red contour line (Z= 0m) indicates the limit of Qattara Depression. There are also marked two highlands of Z= 120m (red contour lines) between of which passes the*
*narrow passage (two black arrows). The drainage divide is marked with a black dotted line (Diffa Plateau). The proposed exit of the Argonauts from Qatarra Depression to the Mediterranean Sea (black line) passes through several deirs depressions), which their dimensions declare the existence of old lakes. There are marked on the map with blue color.*

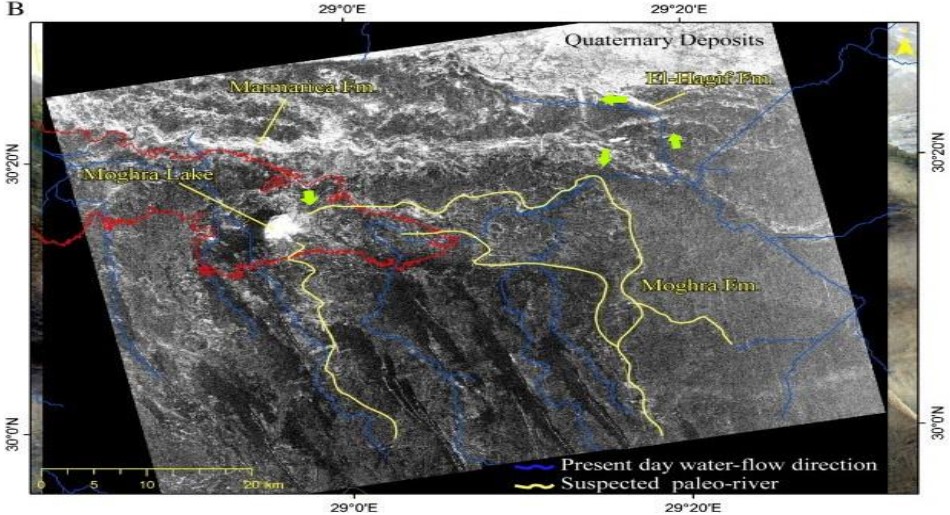

*Figure 10. Optical and radar remote sensing images by Khan et al (2014). PALSAR image showing shallow subsurface fluvial systems (yellow lines). Present-day streams are also shown (blue lines), extracted from SRTM 90-m data. The proposed Argo's route coincides with one of these streams (green arrows).*

Of course, we don't have enough geologic data, such as studied and dated cores from drillings, geomorphologic analyses, or hydrographic data to deduce a safe result. In a study of this kind, there is always a degree of uncertainty. However, Apollonius was found correct in all of his narrated details up to the narrow passage he describes. For this reason, we consider that the percentage of uncertainty is very small. Additionally, *based on the details of the ancient text*, we found (section 1) that Argo's 'arrival point' from Tritonis Lake was close to El Alamein, in the Egyptian coast. In this section, *following an absolutely different method*, we found that the exit of Tritonis Lake was close to El Alamein, which is exactly the same point produced by our former different analysis.

Finally, the entire route of the Argo, from the Major Syrtis Gulf to the Tritonis Lake (Siwa Oasis and Qattara Depression) and its exit from it, through a narrow passage, to the coast, near to El Alamein is presented in Fig. 7 (bottom).

**Conclusions**

The existence of Tritonis Lake, somewhere in North Africa, is connected with the Argonautic Campaign which must have taken place some decades before the Trojan War, at about 1300 BC. Argo reached to this lake, coming from Syrtis Gulf, and through it, the ship, from the African coast, arrived to East Crete and Anafi Island. This route is based on both ancient texts (*Orphica Argonautica* and *Apollonius' Argonautica*) and the *archaeological findings* of the temple of Apollo, on Anafi Island. However, the exact location of Tritonis Lake was unclear because the lake had been dried up during the *dark ages*. There is a disagreement of the authors (ancient and modern), as to the exact location of the lake. According to Apollonius of Rhodes, the Argonauts carried the ship from Syrtis Gulf, for 12 days and nights, in *the desert of Libya inland* where they arrived in Tritonis Lake. Also, Apollonius writes that Triton is the oldest name on the Nile River (*Argonautica, IV, 259-260 &267-271*).

In this work, following step by step Apollonius' description, we found the pre-historic Tritonis Lake's location *in the water environment of Siwa Oasis and Qattara Depression*, in Egypt. We accept that the lakes in Siwa Oasis communicated with each other and Qattara Depression's environment, in the wet conditions of the 13th century BC. There was a 'water continuation' between Siwa Oasis and part of Qattara Depression in today's *Dabaa formation* and the *sabkha formations*, allowing sailing within it, from south to north.

We ruled out the possibility that the Tritonis Lake was located near Minor or Major Syrtis or in Cyrenaica peninsula, as proposed by various (ancient and modern) authors, for two significant reasons: a) in all these cases, Argo had to approach *Cyrene peninsula* and from there to reach *east Crete*. However, Apollonius describes that the Argonauts, sailing away from the African coast, *saw from afar first the 'mountains of Karpathos Island'* and then the mountains of Crete (IV, 1635-1640).



Karpathos is a small island which is located northeast of East-Crete. It is obvious that if a ship travels from any location of Western and Central African coast (up to Cyrene) to Crete, its crew will first see the high mountains of Crete. This Apollonius' narrative is only fulfilled if the place from which Argo departed was located *east of Marsa Matruh*, meaning in Egypt. b) Cyrene has a distance of only 305 km from *western Crete* (Palaeochora port, Chania). The Argonauts *would go directly northwards from Cyrene to west Crete* and from there they would travel *easily* to Peloponnesus, which was the final

destination of their travel in order to continue their trip northwards to Iolcus. The distance between Chania (west Crete) and Neapoli (Maleas Cape, in south Peloponnesus) is only 141.4 km. Even in the case that the ship pushed to east Crete, accidentally, by the change of the wind from south (*as Apollonius described*) to south-eastern direction, it would be logical for the Argonauts to follow a coastal route along south Crete going *westwards* in order to approach Peloponnesus. There was no reason for them to risk themselves in the Cretan sea, sailing to eastern Crete and *then turning back again, westward*, in

order to approach the Peloponnesus.

    The Argonauts transferred the ship from Major Syrtis Gulf and following *the traces of Poseidon's horse* reached to Tritonis Lake, according to Apollonius. These 'traces' are *the remnants of the older waters*, in the form of shallow lakes or rivers (e.g. today's sabkha, wadi etc), one after the other, existing in the wet period of 13th century BC, in the north edge of the Libyan Dessert. This transformation is done sometimes by floating the ship in the existing waters and sometimes by pulling

and dragging the ship on the sand, using the strength of their shoulders. The know-how of this movement was known to them. This route until Siwa Oasis has a length of 633 km and was realized with an *average speed* of 2.2 km/hour, nonstop (that is the transportation was in double guard), during 12 days and nights.

    The Mediterranean Sea is exactly northwards from the north edge of Qattara Depression. Its northern edge consists of steep hills and there is not any obvious water corridor from this place towards the Mediterranean Sea. Indeed, according to

Apollonius, the outlet of Tritonis Lake is described as a *narrow passage*. On both sides of it, there are steep rocky exposures exactly like the ones of the hills in the Qattara Depression. According to our calculations, this *narrow water passage* started from Moghra Lake (north-east of Qattara Depression), traversed the Diffa Plateau (northern limit of Qattara Depression) and ended at *Sidi Abd El Rahman, located 20 km west of El Alamein,* on the Egyptian coast. Today, this route is covered by traces of ancient lakes and ancient rivers.

From there, according to Apollonius, Argo sailed eastwards for 24 hours and reached at a promontory from which the ship moved away from the African coast, towards east Crete. This location of the Argo's departure is the *Abu Qir cape,* in the Canopis branch of Nile Delta. This connection between *Argo and Canopus Delta* occurs in the night sky. The brighter star of the constellation Argo is named Canopus. The ancient Greek writers who were giving names to the stars and constellations, much later, they deemed it appropriate *to link* Argo's arrival in Canopus Delta, because this place was crucial for the

Argonauts' return.

    Apollonius identified the '*garden of Hesperides or the Gods' garden*' within the pre-existing field of Atlas ('*Atlas' farm*'), near to the *entrance of Tritonis Lake* and he characterized this place '*sacred plain*'. However, the sanctity of Siwa Oasis is a

given, because of the ancient sacred oracle of Ammon-Zeus. We have proved that the original oracle existed *long before* arrival of Argonauts there. The construction of the oracle is such that sunlight illuminates exactly its center, during the

winter solstice (then, the zodiac sign of Capricorns). Gregory and Gregory (1999) referring to Richer (1989) *geodetic network work*, note that 'all decorations …of the temples had a specific meaning by which an observer located *at the temple center* had to see ..on the walls …a mythological scene that was associated with the God *related to the zodiac sign*….In this way, *every traveler who needed orientation*, had to search for a temple and from its dedication he immediately knew his azimuthally location with respect to the 'navel'. In this geodetic system, the oracle of Ammon-Zeus and the sacred island of

Delos are located in the same meridian and their distance bisects the distance between Delphi and Sardis oracles.

Ammon-Zeus, who was a solar deity, bears ram's horns ('*kerasforos*'), because he is connected with, *then,* the zodiac sign of the Spring Equinox which was the Aries (*ram*). This zodiac sign represents, mythological, the '*golden ram'* whose *Golden Fleece was* transferred by Argo and was on the ship, when it reached Siwa Oasis. According to Quintus Curtius and Diodorus Siculus, in this oracle there was a very old ritual. Eighty priests carried a sacred stone or xoanon of conical shape

(named 'navel'), which 'symbolized' Ammon-Zeus, upon a 'golden boat', ceremoniously. It is noticeable that the priests were transporting the *'symbol of the god, the navel'* and not the *'image (e.g. statue) of the god himself'*. Is it possible this ritual mimic the unexpected appearance of the ship (Argo) with the *Golden Fleece (and not the golden ram itself)* on it, in Siwa Oasis?

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
