# Peer review of "A New Propose for Prehistoric Tritonis Lake's Location based on Apollonius of Rhodes' description"

_History of Geo- and Space Sciences, 2022_

## Referee Comment (RC1)

This paper made me realise how little I knew about the Argonauts and their journey. I knew about the dragon's teeth. I knew about the golden fleece. I knew about Jason. I was unaware that Hercules (or is it Heracles?) was involved in all of this. And I was certainly unaware that this was all written down by Apollonius of Rhodes.

I gather that Apollonius is believed to have lived from 300 to 225 BC. Yet, as this paper points out (line 50) the Argonauts sailed in the 13th century BC. The authors of this paper are therefore trying to determine events and locations on the basis of a text written 1000 years after the events. This allows the authors the latitude to claim that certain parts of the text are exact, if they agree with their arguments, but that other parts are inexact or poetic if they disagree with their arguments. Such latitude in interpretation does not sit well with scientists.

In particular, the authors like to claim in lines 234, 370, 390, and 682 that they have "proved" certain things. They may indeed have shown, demonstrated or even confirmed their theories but their claims to have proved them diminishes their credibility. Dogmatic statements such as "Herodotus would certainly refer to that as well" (line 115) need further explanation. Herodotus does not have a good reputation as an accurate source of information (my wife is a historian), and I quote from the internet:

> Herodotus has been called both the Father of History and the Father of Lies. Although his Histories are our primary source for knowledge of the Persian invasions of Greece in 490 and 480 B.C., Herodotus' account includes some elements which seem to us incredible, sometimes almost bordering on the realm of fantasy.

The authors claim to have determined the location of the Triton's Lake, or Tritonis Lake as they call it. They may have. Or they may not have. There seems no way of scientifically determining this. In which case my judgement on this paper rests on whether it is interesting, and whether it reads well. In short, the answer to question 1 is yes. The answer to question 2 is no.

Interest

When I was a youngster there was an Argonauts' club on the radio (there was no TV back then). The Argonauts' club subtly introduced generations of Australian children to quality literature, songs, music and games on the ABC broadcasting service daily from 5pm to 6pm. Ostensibly the children's hour, at some point during the hour the Argonaut theme song would be played and the jolly friendly compere known as Mac would turn into the stern Jason. When one joined and became an Argonaut one supposedly joined Jason and his vessel *Argo*, setting out to find the Golden Fleece and was assigned a boat and a rowing position that became one's Argonaut Club name. There were fifty 'rowers' in each 'boat'. I was Boronia 18. One received certificates for writing about the name of the boat. Despite consulting the Encyclopedia Britannica in the Lindfield library, run by the local council, I could find nothing about Boronia. There is a genus of Australian plant called Boronia, but as far as I can make out neither the plant nor the chemical element Boron got its name from any Greek word. The closest Greek word is boras, meaning north, as in the aurora borealis and I suspect that the literary types running the club were running out of names by this time and picked anything sounding vaguely Greek. Had I known about Tritonis Lake and written about it then I am sure that I would have received lots of certificates and perhaps even attained the Dragon's Tooth badge, or the Golden Fleece badge.

Obviously I have a soft spot for stories about the Argonauts and now realise how little I know.

Readability

The paper is lacking in many areas. Some that I noted are:

1. Background: I suspect that most readers will not have any knowledge about the Argonauts, nor about the poems on which this analysis is based. The background information that is given is insufficient with one glaring omission being a good, readable map that shows all the places mentioned in the paper. I had to read this paper with an atlas open at the page showing the Mediterranean and even then I had difficulty. My atlas, for example, does not (I think) show that "The Minor Syrtis is in Tunisia, near to the Mountain Atlas' area "(line 80).

   To further confuse the poor uninformed reader, the authors refer to 'both Syrtis' (line 69) long before Syrtis Minor has been mentioned.

2. Poor English: The English in the paper is just marginally adequate. It is below the level expected in an English language scientific publication. Some of the numerous examples follow:

   The title starts with "A New Propose…". Propose is a verb. The word should be 'Proposal' though words such as Idea or Concept may be more suitable.

   Line 17 has Apollonius Rhodes whereas the title has Apollonius of Rhodes

   Line 28 Colloquial English would omit 'the' in both places on this line.

   Line 29 'Triton who' not 'Triton which'.

   Line 30 'to have' should be either 'as having' or 'with'

   Line 41 'toddler bids and he farewell' should be 'toddler and he bids farewell'.

   Line 53 'lake has been dried up' should be 'lake dried up'; 'authors have difficulties to agree to it location' should be 'authors fail to agree as to its location'.

   Line 61 'taken' should be 'took'.

   Line 104 'clarify' should be 'clarifies'.

   There are lots more and I am sure that the authors will have fun finding them.

   For reasons that defy comprehension, when English uses the Latin Major and Minor they are placed after the noun so that it would be more usual to refer to Syrtis Major and Syrtis Minor.

3. Obscure Meaning. This may be my fault but I think that words such as trans-scientifically; oasis; sabkha; and thalweg should be defined or explained. I also wonder if there is a

difference between a sabkhat (line 330) and a sabkha.

I do not understand what a 'reciprocal coastal wave' is (line 29), which is embarrassing as I thought that I knew something about coastal waves, having read Mysak & LeBlond's book "Waves in the Ocean".

Dark Ages (line 62). I am fairly certain that in English, the term Dark Ages is used for the period 476 AD – 1000 AD, and only for that period. To use it for a time period in the 12[th] century BC is very confusing.

A similar problem exists with the use of 'West and Central African Coast' (lines 159 and 181). West Africa and Central Africa are both in the Gulf of Guinea. The authors mean the western and central portions of the North African coast (of the Mediterranean).

The numbering of the Figures does not agree with the text. There is, for example, no Figure 6.

4. Inconsistent or obscure referencing.
   4a. Though Appolonius and Eratosthenes appear in the list of references, the following ancient sources that are mentioned in the paper do not:
   Pausanias, Hecataeus, Orpheus, Homer, Herodotus, Pindarus, Strabo, Dionysius Schytovrachion and Diodorus Siculus and possibly others.

   4b. In the reference list itself, the year is generally at the end except that in lines 707 and 747 it appears after the authors names.

   The reference list has 2002 for Aref et al, but line 453 has 2001. Which is curious since line 454 has 2002. Unclear if there is a missing reference or whether it is just a mistake.

   The Journal names are sometimes given in full. Sometimes not. I am not sure how many readers will know: MAA, Q Sc R, Am.Ph.As, J. Ar. Sc.,

5. Relevance to Geophysics
   In line 23 the reader is informed that 'modern technology' was used. If I understand the paper correctly, the modern technology consisted of a digital elevation model determined using satellite data that was then used to infer the pre-historic thalwegs. I suggest changing 'modern technology' to 'a digital elevation model'.

   I think that claiming all of this to be "significant to geophysical research" (line 124) is gross exaggeration.

---

## Community Comment (CC3)

[revised manuscript text omitted]

---

## Community Comment (CC7)

[revised manuscript text omitted]

---

## Author Comment (AC1)

Answer

Dear Mr. Beer,

We are glad that our paper enlarged your knowledge in connection with the Argonautic Campaign. It seems that your interest starts from your young years from the Argonauts' club of Australia.

In your text you have sent to the site of the journal you express a doubt about our ideas expressed in the paper. Initially, you assume that Apollonius' text cannot be correct since 1000 years have been elapsed since the events. I would like to inform you that Apollonius was director of the famous library of Alexandria in which all the wealth of the ancient knowledge was available to the researchers there. Consequently he could have many and serious primary sources.

In the paper 'Voett A., Bruckner H., Schiever A., Handi M., Besonen M. and Van der Borg K. (2004): Holocene coastal evolution around the ancient sea port of Oiniadai, Acheloos alluvial plain, N.W. Greece. In Schernewski, G., Dolch, T . (Eds) Geographie der Meere und Kunsten. Coastline Reports, vol. I, pp.43-53, Rostock-Warnemunde, you can recognize the evolution versus time of this region after the start of the melting of the last ice age from 18000 before present and onwards. Ovidius a Latin author, in Metamorphoses, VIII, 576 described an old myth which exhibited a cataclysmic event occurred in West Greece involving, not 1000 years distance from him, as it was the case of Apollonius, but several thousands of years distance from him. What he described has been proven fully as the following maps illustrate clearly. Mariolakos et al., 2017. Proceedings of the International symposium 'Ancient Greece and the modern world: Ancient Olympia 28-31 August in 2016. p.p. 299-322.

The geological studies dated the gradual raising of the sea level. The local population, there, conceived the river deposits of the proto-Acheloos river as Nymphs of the river and the latter as a 'god'. In the following Figures the evolution of the coast lines is presented. The studies dated the sea event versus time and they explained it fully validating Ovidius.

[Figure]

In these Figures you can see successfully our understanding of the coastal changes of West Greece in which the river deposits became islands due to raising of the sea's level which was

a result of the melting of the glaciers of the last ice age. The old myth wanted 'god' Acheloos to be angry because his nymphs did not respect him anymore. Time has passed and Zeus and the Olympians were in action. Therefore he took the nymphs and their land into the sea! The nymphs today are called Echinades and they are islands

Your comment: ' this allows the authors the latitude to claim that certain parts of the text are exact, if they agree with their arguments, but that other parts are inexact or poetic if they disagree with their arguments. Such latitude in interpretation does not sit well with scientists', has nothing to do with the reality. On the contrary, in all the flow of our paper we give emphasis to the reader that we follow the ancient text very carefully and we demonstrate the absolute agreement between our findings and the writings of Apollonius. We never wrote that we found disagreement between our findings and the ancient text or that we accepted 'poetic permission'. Where did you see that in our paper?

It does not make sense your remark 'their claims to have proved them diminish their credibility', since we have proved, with all the information, we supply, our conclusions. If we did not produce the proofs you could write about lack of credibility. But we have supplied all the evidence of our findings in a stepwise mode. Therefore your *statement* is out of *time* and *space.*

You write: 'dogmatic statements' isolating one only phrase from an entire paragraph. This phrase is our comment to a Herodotus' text.  Herodotus does write for another Argo's voyage. In that other voyage Argo reached to Minor Syrtis. The historian does not mention the trip of the Argonautic Campaign. See our relative comment in our paper. Consequently we judged to give emphasis and set this difference to the international reader and say that if the Argonauts had arrived for second time to Minor Syrtis during the Argonautic Campaign, Herodotus would have reported it.

Your comment about Herodotus may lead you in the complete rejection of his entire work. It would be a great scientific mistake.

Not all modern historians view the same attitude with respect to Herodotus as a historian. Especially, those, who are informed from the scientific contributions from other realms of sciences such as Geophysics.  For instance the following contribution demonstrated very clearly, to the international scientific community, that Herodotus was absolutely exact in all he narrated in connection with the Xerxes' Canal.

We offer you geophysical literature in which we demonstrated, very clearly, how we tested Herodotus in East Chalkidiki peninsula in Greece and we found him exact at all points he stated.

Karastathis, V.K. and Papamarinopoulos, St.P., (1997). The detection of the King Xerxes' Canal by the use of shallow seismic reflection seismic. Preliminary results. Geophysical Prospecting, 45, 3899-401.   (The paper presented in the European Geophysical Union and was characterized the very paper of the symposium)

Karastathis, V.K., Papamarinopoulos, St.P. and Jones, R.E., (2001) 2-D velocity of the buried ancient canal of Xerxes: an application of seismic methods in archaeology, Journal of Applied Geophysics 47, p.p. 2943.

The canal in East Chalkidiki peninsula is visible exhibiting all the measurable parameters described by Herodotus in his text.  (Herodotus Histories 7.23.8 7.23.29).

[Figure]

We have used advanced geophysics in order to test Herodotus and we found him correct at all aspects.

I would like to inform you that in the original ancient Greek text the lake is called Tritonis and not Triton's lake. Consequently your comment 'Tritonis lake as they call it, meaning us, is not correct. The naming of the lake was not given by us.

You judge the paper interesting but it is not well written. You had the kindness to show us some points of improvement. We shall consider them after receiving the answer from the reviewers.

For instance we could add a map in the Appendix as you suggest. We could add a glossary in the end if the reviewers ask it and the journal allows some extra pages.

 By the way, Minor Syrtis is in Tunisia, is the Gulf of Gabes.

Figure 6 is mentioned in our text normally in the line 475.

The Dark Ages of Greece have nothing to do with other Dark Ages of other countries. *I thank* you for your remark and for the *correction* of the phrase 'West and Central African Coast'.

We shall take account your remarks in connection with the bibliography.

The 'modern technology' could be written indeed 'digital elevation model'.

We also used *forward and reverse engineering* in order to locate the departure's point of the Argonauts to sail from the African coast initially within the Mediterranean and eventually to reach Aegean Sea.

Our paper offers useful information for the hidden water sources to the geophysical science in the form of detailed aeromagnetic, aerogravimetric and aeroradar mapping supplying data for further researches for the tectonic regime and the unknown hydrogeological environment of North East Africa for which the water resources will be needed badly in the near future.

 Our paper contributed in defining an important astrogeodetic point located in Siwa Oasis, the understanding of the ancient cartography within North East Africa defining an unknown water exit towards the Mediterranean Sea, the understanding of ancient maritime voyages connecting the Aegean Sea and Africa.  This is not a *gross exaggeration* but a simple and plain truth.

Prof.Dr. Stavros P. Papamarinopoulos

University of Patras-Lab of Geophysics

Lead Author, HG SS Topical Editor

---

## Author Comment (AC2)

Dear Dr. Tom Beer

Thank you very much for your prompt answer.

About Fig.6: It is located in the bottom of page 19, but its legend is in the top of page 20.

We have not any disagreement with you in connection with the word 'prove'. In all the text of our work we use very careful phrases. Our position is expressed with clarity in line 622-623: *'Of course, we don't have enough geologic data, such as studied and dated cores from drillings, geomorphologic analyses, or hydrographic data to deduce a safe result. In a study of this kind, there is always a degree of uncertainty'.*

You can see the style of our writing which is in agreement with your position in phrases as:

In the last paragraph of our introduction: *'In this work we will try to answer where exactly Tritonis Lake was located based on the detailed description of Apollonius Argonautica text.'*

Line 318: *'We are forced to accept that the Agronauts arrived to the ancient Tritonis Lake which was located in the cluster of lakes in Siwa Oasis'.*

Line 322: *'Thus, we conclude that these 'traces' must be the water route existing in the 13$^{th}$ century BC'*

LINES 344-346: *'We can deduce that in the circumstances in the 13$^{th}$ BC, the same percentage of 70% would have significantly more water allowing Argo's sailing. This percentage increases to more than 70%, due to the possibility that the waters were also covering additional areas'.*

LINES 467-469: *'The increased of the water flows in the 13$^{th}$ century BC (see in Fig. 2 (right)), forces us, to accept the existence of satisfactory volume of water in the Dabaa formations and in the sabkhas which allowed Argo to sail through them'.*

LINE 555: *'…and probably had water in the 13th century BC'.*

LINE 574-575: *'Because they are large enough, could have had significant amounts of water during the 13$^{th}$century BC, just before the 'dry period'. It is very likely that some of them represent palaeo-lake locations'.*

To avoid any misunderstandings, we are going to delete the word 'prove', as you proposed as follows:

Line 234: *'We have proved…'* changes into *'We have concluded based on the Apollonius text..'*

LINE 370: *'A proof that…'* changes into *'a serious indication that..'*

LINE 391: *' Is also proved by…'* changes into *' is also arises from…'*

LINE 683: *'We have proved that …'* changes into *'It is known that…'*

The updated text of our paper including your remarks and corrections will be sent to you as soon as possible.

Sincerely yours

Professor Stavros Papamarinopoulos

---

## Author Comment (AC4)

Dear Prof.T.Beer,

I would like to thank you for your kindness in reading our paper in connection with the Argonauts. We have included all of your remarks in the second version of the paper. Moreover, according to your instructions, we followed your guidance and produced a glossary and a map in the Appendix, showing all the positions of the places related to the southern Argonautic campaign. We also resolved all the issues regarding the bibliography section, as you requested.

The paper is almost ready and prepared to be submitted to the editors. Furthermore, we are going to include interesting remarks from other colleagues who have also read the paper from the preprint file.

Always with health,

Prof. Dr. Stavros Papamarinopoulos

Lead author

HG SS Topical Editor

---

## Author Comment (AC8)

Dear Professor Cathcart,

We would like to thank you for your professional review of our paper and for your interest and your guidance regarding its content.

  (1) A title change seems evident: it's a new "proposal" based on records so it is NOT "prehistoric".

Thanks to your suggestion we change the title as follows (see also our previous answer):

'A new trip itinerary locating Tritonis Lake and the unknown water route, north of it, which led the Argonauts back to the Aegean Sea, based, on Apollonius of Rhodes text'.

  (2) Overlooked is the aspect of "sand surfing" those Argonaut boats. It is a desert-area sport in California, for example.

The wind could have been an enhancing factor in the whole effort as the ship had sails, especially when it was in a water environment. Additionally, the ancient Greek text denotes the existence of readymade available rollers in order to tow the ship in any case where this was needed.

*(3)* Could the exit route from Tritonis have been, in part, and overland roller-assisted portage like the one at Corinth before the canal was dug?

Indeed, the 18-km-long 'narrow passage' could easily be used with a diolkos system of rollers, as in ancient Korinthos during the classical period of Greece. However, the system of rollers also existed in the prehistoric period. For the 'narrow passage' in lines 601-610, we propose, that the narrow passage, was possibly artificial, since the prehistoric Egyptians had constructed such works much earlier than the 13th century B.C. *"Even if there was no water in this gorge in the 13$^{th}$century BC, Egyptians possessed the technology to build a channel, taking advantage of the existence of the fault, in order to create a water route from the old Tritonis Lake to the Diffa Plateau and eventually to the Mediterranean Sea".*

By the way, the chapter you wrote in the volume edited by Viorel Badescu Richard B. Cathcart Editors, 'Macro-engineering Seawater in Unique Environments Arid Lowlands and Water Bodies rehabilitation' is of great interest because it fits fully with the arid environment in which the Argonauts were operating in the 13$^{th}$ BC. We intend to include it in the bibliography of our work.

---

## Author Comment (AC9)

Dear Professors and Doctors
Tom Beer (reviewer 1)
Richard  Cathcart
Charles Finkl
Niki Evelpidou

This email is addressed to you who were kind enough to study our work. Your remarks and the suggested corrections helped us to improve the manuscript.

We consider as necessary to send you the new and updated version of our manuscript. In yellow and green color we tracked the changes based on your way of thinking. If you think that further corrections are needed, we are open to review the text once again. But this procedure should be carried out within a week if possible.

This email is forwarded to both the Editorial team and the Secretary of the journal. We would like to inform you that the publication of our paper is pending for several months due to the fact that no second formal reviewer has been nominated by the editorial team, as the latter, invariably informs us every two weeks!

In spite, of the fact, that the initial paper has been reviewed 'four times all of by you', we are wondering if there is a way in order the editorial team to be assisted in nominating the 'second reviewer', because it continues attempt unsuccessfully to nominate another reviewer.

With regards

The Authors team:

Prof. S. Papamarinopoulos (University of Patras, Greece), member of the Editorial team of this magazine
Prof. H. Maroukian (University of Athens, Greece)
As. Prof. P. Preka-Pamadema (University of Athens, Greece)
As. Prof. Ch. Tzanis (University of Athens, Greece)
K. Kalachanis (Researcher, New York College, Athens, Greece)
G. Sarantitis (Researcher)
D. Theodosopoulos (Researcher)